# Dynamically evolving novel overlapping gene as a factor in the SARS-CoV-2 pandemic

Chase W Nelson[1,2†]*, Zachary Ardern[3†]*, Tony L Goldberg[4,5], Chen Meng[6], Chen-Hao Kuo[1], Christina Ludwig[6], Sergios-Orestis Kolokotronis[2,7,8,9], Xinzhu Wei[10,11]*

[1]Biodiversity Research Center, Academia Sinica, Taipei, Taiwan; [2]Institute for Comparative Genomics, American Museum of Natural History, New York, United States; [3]Chair for Microbial Ecology, Technical University of Munich, Freising, Germany; [4]Department of Pathobiological Sciences, University of Wisconsin-Madison, Madison, United States; [5]Global Health Institute, University of Wisconsin-Madison, Madison, United States; [6]Bavarian Center for Biomolecular Mass Spectrometry (BayBioMS), Technical University of Munich, Freising, Germany; [7]Department of Epidemiology and Biostatistics, School of Public Health, SUNY Downstate Health Sciences University, Brooklyn, United States; [8]Institute for Genomic Health, SUNY Downstate Health Sciences University, Brooklyn, United States; [9]Division of Infectious Diseases, Department of Medicine, SUNY Downstate Health Sciences University, Brooklyn, United States; [10]Departments of Integrative Biology and Statistics, University of California, Berkeley, Berkeley, United States; [11]Departments of Computer Science, Human Genetics, and Computational Medicine, University of California, Los Angeles, Los Angeles, United States

*For correspondence:
cnelson@amnh.org (CWN);
zachary.ardern@tum.de (ZA);
aprilwei@berkeley.edu (XW)

[†]These authors contributed equally to this work

Competing interests: The authors declare that no competing interests exist.

**Abstract** Understanding the emergence of novel viruses requires an accurate and comprehensive annotation of their genomes. Overlapping genes (OLGs) are common in viruses and have been associated with pandemics but are still widely overlooked. We identify and characterize *ORF3d*, a novel OLG in SARS-CoV-2 that is also present in Guangxi pangolin-CoVs but not other closely related pangolin-CoVs or bat-CoVs. We then document evidence of *ORF3d* translation, characterize its protein sequence, and conduct an evolutionary analysis at three levels: between taxa (21 members of *Severe acute respiratory syndrome-related coronavirus*), between human hosts (3978 SARS-CoV-2 consensus sequences), and within human hosts (401 deeply sequenced SARS-CoV-2 samples). *ORF3d* has been independently identified and shown to elicit a strong antibody response in COVID-19 patients. However, it has been misclassified as the unrelated gene *ORF3b*, leading to confusion. Our results liken *ORF3d* to other accessory genes in emerging viruses and highlight the importance of OLGs.

## Introduction

The COVID-19 pandemic raises urgent questions about the properties that allow animal viruses to cross species boundaries and spread within humans. Addressing these questions requires an accurate and comprehensive understanding of viral genomes. One frequently overlooked source of novelty is the evolution of new overlapping genes (OLGs), wherein a single stretch of nucleotides encodes two distinct proteins in different reading frames. Such 'genes within genes' compress genomic information and allow genetic innovation via *overprinting* (*Keese and Gibbs, 1992*), particularly

as frameshifted sequences preserve certain physicochemical properties of proteins (*Bartonek et al., 2020*). However, OLGs also entail the cost that a single mutation may alter two proteins, constraining evolution of the pre-existing open reading frame (ORF) and complicating sequence analysis. Unfortunately, genome annotation methods typically miss OLGs, instead favoring one ORF per genomic region (*Warren et al., 2010*).

OLGs are known entities but remain inconsistently reported in viruses of the species *Severe acute respiratory syndrome-related coronavirus* (subgenus *Sarbecovirus*; genus *Betacoronavirus*; *Coronaviridae Study Group of the International Committee on Taxonomy of Viruses et al., 2020*). For example, annotations of *ORF9b* and *ORF9c* are absent or conflicting in SARS-CoV-2 reference genome Wuhan-Hu-1 (NCBI: NC_045512.2) and genomic studies (e.g. *Chan et al., 2020*; *Wu et al., 2020b*), and OLGs are often not displayed in genome browsers (e.g. *Flynn et al., 2020*). Further, *ORF3b*, an extensively characterized OLG within *ORF3a* that is present in other species members including SARS-CoV (SARS-CoV-1) (*McBride and Fielding, 2012*), has sometimes been annotated in SARS-CoV-2 even though it contains four early STOP codons in this virus. Such inconsistencies complicate research, because OLGs may play key roles in the emergence of new viruses. For example, in human immunodeficiency virus-1 (HIV-1), the novel OLG *asp* within *env* is actively expressed in human cells (*Affram et al., 2019*) and is associated with the pandemic M group lineage (*Cassan et al., 2016*).

## Results

### Novel overlapping gene candidates

To identify novel OLGs within the SARS-CoV-2 genome, we first generated a list of candidate overlapping ORFs in the Wuhan-Hu-1 reference genome (NCBI: NC_045512.2). Specifically, we used the codon permutation method of *Schlub et al., 2018* to detect unexpectedly long ORFs while controlling for codon usage. One unannotated OLG candidate, here named *ORF3d*, scored highly (p=0.0104), exceeding the significance of two known OLGs annotated in Uniprot (*ORF9b* and *ORF9c*, both within *N*; https://viralzone.expasy.org/8996) (*Figure 1*, *Figure 1—figure supplement 1*, and *Supplementary file 1*).

*ORF3d* comprises 58 codons (including STOP) near the beginning of *ORF3a* (*Table 1*), making it longer than the known genes *ORF7b* (44 codons) and *ORF10* (39 codons) (*Supplementary file 1*). *ORF3d* was discovered independently by *Chan et al., 2020* as '*ORF3b*' and *Pavesi, 2020* as 'hypothetical protein'. Due to this naming ambiguity and its location within *ORF3a*, *ORF3d* has subsequently been conflated with the previously documented *ORF3b* in multiple studies (e.g. *Fung et al., 2020*; *Ge et al., 2020*; *Gordon et al., 2020*; *Hachim et al., 2020*; *Helmy et al., 2020*; *Yi et al., 2020*). Critically, *ORF3d* is unrelated (i.e. not homologous) to *ORF3b*, as the two genes occupy different genomic positions within *ORF3a*: *ORF3d* ends 39 codons upstream of the genome region homologous to *ORF3b*, and the *ORF3b* start site encodes only 23 codons in SARS-CoV-2 due to a premature STOP (*Wu et al., 2020a*; *Table 1*, *Figure 1*, *Figure 1—figure supplement 1*, and *Supplementary file 1*). Furthermore, the two genes occupy different reading frames: codon position 1 of *ORF3a* overlaps codon position 2 of *ORF3d* (frame ss12) but codon position 3 of *ORF3b* (frame ss13). *ORF3d* is also distinct from other OLGs hypothesized within *ORF3a* (*Table 1*). Thus, *ORF3d* putatively encodes a novel protein not present in previously discovered *Severe acute respiratory syndrome-related coronavirus* genomes, while the absence of full-length *ORF3b* in SARS-CoV-2 distinguishes it from SARS-CoV (*Figure 1*).

### *ORF3d* molecular biology and expression

To assess the expression of *ORF3d*, we re-analyzed the SARS-CoV-2 ribosome profiling (Ribo-seq) data of *Finkel et al., 2020*. This approach allows the study of gene expression and reading frame at single nucleotide (nt) resolution by sequencing mRNA fragments bound by actively translating ribosomes (ribosome footprints) (*Ingolia et al., 2009*). To do this, we focused on samples with reads reliably associated with ribosomes stalled at translation start sites, i.e. lactimidomycin and harringtonine treatments at 5 hr post-infection. After trimming reads to their 5′ ends (first nt), we observe a consistent peak in read depth ~12 nt upstream of the start site of *ORF3d* (*Figure 2A*), the expected

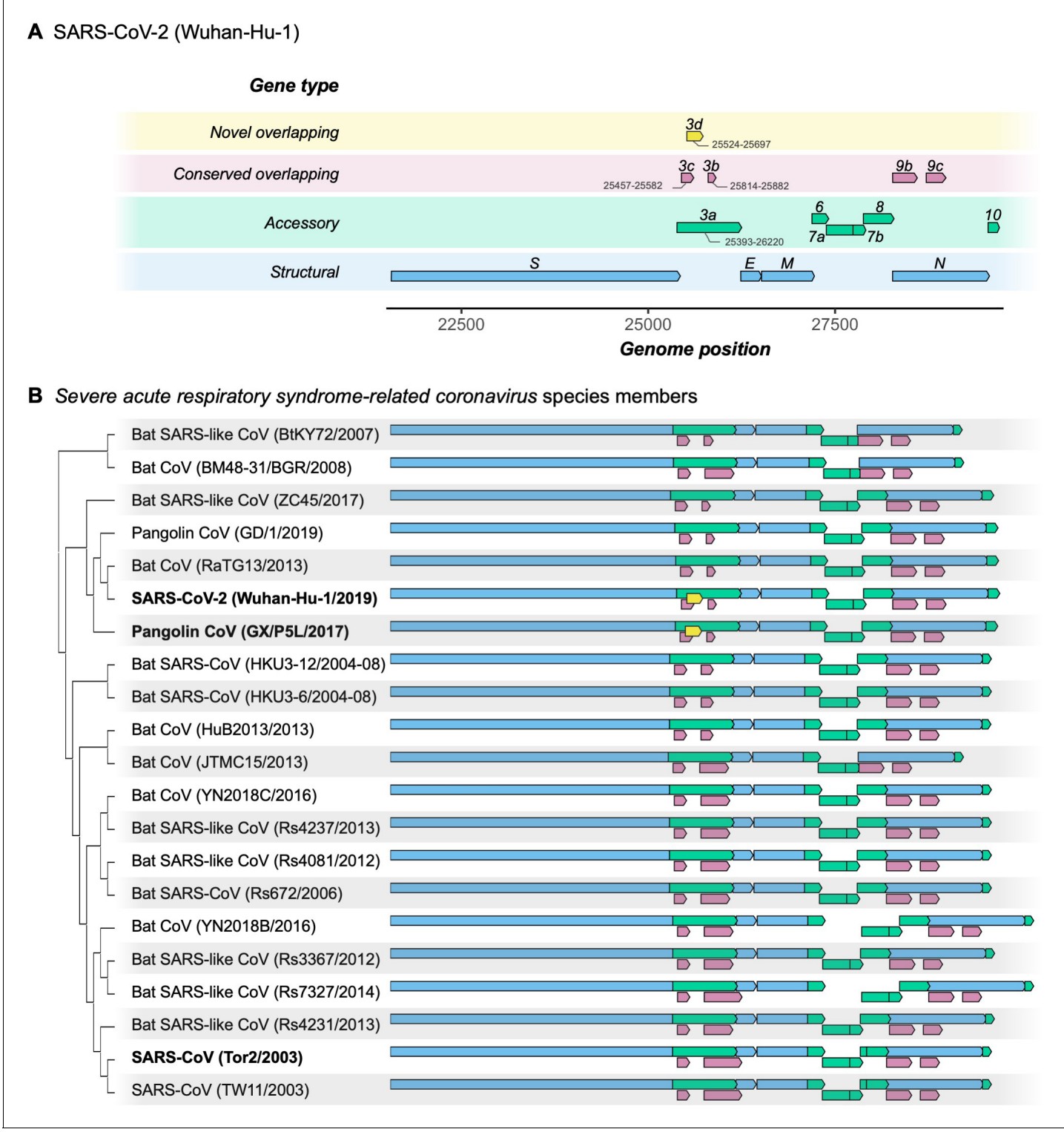

**Figure 1.** Gene repertoire and evolutionary relationships of *Severe acute respiratory syndrome-related coronavirus* species members. Only genes downstream of *ORF1ab* are shown, beginning with the Spike gene *S*. (**A**) Four types of genes and their relative positions in the SARS-CoV-2 Wuhan-Hu-1 genome (NCBI: NC_045512.2). Genes are colored by type: novel overlapping genes (OLGs) (gold; *ORF3d* only); conserved OLGs (burgundy); accessory (green); and structural (blue). Note that *ORF3b* has been truncated relative to SARS-CoV genomes, whereas *ORF8* remains intact (i.e. has not been split into *ORF8a* and *ORF8b*). (**B**) Genes with intact ORFs in each of 21 *Severe acute respiratory syndrome-related coronavirus* genomes. Gene positions are shown relative to each genome, i.e. homologous genes are not precisely aligned. Only the full-length isoforms of *ORF3a* and *ORF3d* are shown (for shorter isoforms, see *Table 1*). Note that the first 20 codons of *ORF3d* overlap the last codons of *ORF3c* (*Supplementary file 1*), such that

*Figure 1 continued*

the beginning of *ORF3d* involves a triple overlap (*ORF3a*/*ORF3c*/*ORF3d*). *ORF3b* is full-length in only three sequences (SARS-CoV TW11, SARS-CoV Tor2, and bat-CoV Rs7327), while the remaining sequences have premature STOP codons (*Supplementary file 1*). *ORF8* is not novel in SARS-CoV-2 (contra *Chan et al., 2020*), but is intact in all but five sequences (split into *ORF8a* and *ORF8b* in SARS-CoVs TW11 and Tor2; deleted in bat-CoVs BtKY72, BM48-31, and JTMC15). *ORF9b* and *ORF9c* are found throughout this virus species, yet rarely annotated in genomes at NCBI.

The online version of this article includes the following source data and figure supplement(s) for figure 1:

**Source data 1.** SARS-related-CoV_ALN.fasta.
**Source data 2.** SARS-related-CoV_ALN.gtf.txt.
**Figure supplement 1.** Codon permutation analysis to identify candidate overlapping genes in all three forward-strand reading frames of the SARS-CoV-2 genome.

ribosomal P-site offset (*Calviello and Ohler, 2017*). Similar peaks are also observed for previously annotated genes (*Figure 2A*, *Figure 2—figure supplement 1*).

To investigate the relationship between ribosome profiling read depth and expressed protein levels, we re-analyzed five publicly available SARS-CoV-2 mass spectrometry (MS) datasets (*Bezstarosti et al., 2020*; *Bojkova et al., 2020*; *Davidson et al., 2020*; PRIDE Project PXD018581; *Zecha et al., 2020*; Materials and methods). We were unable to detect ORF3c, ORF3d, ORF3b, ORF9c, or ORF10 when employing a 1% false-discovery threshold. This result may reflect the limitations of MS for detecting proteins that are very short, weakly expressed under the specific conditions tested, or lack detectable peptides; for example, even the envelope protein E is not detected in some SARS-CoV-2 datasets (*Bojkova et al., 2020*; *Davidson et al., 2020*). However, we do find a strong correlation between protein expression as estimated from MS and the ribosome profiling read depth observed at upstream peaks, with $r_S$ = 0.89 (p=0.0004, Spearman's rank) (*Figure 2*, *Figure 2—figure supplement 2*). This suggests that the presence and depth of an upstream peak is a reliable indicator of expression. Specifically, results from both proteomic and ribosome profiling approaches confirm that N, M, and S are the most highly expressed proteins, with N constituting ~80% of the total viral protein content. We also observe that the number of reads determined to be in the correct reading frame (codon position 1) along the gene is moderately correlated with MS expression values ($r_S$ = 0.60, p=0.056), while this is not true for out-of-frame (codon position 2 and 3) reads ($r_S$ = 0.39, p=0.237) (*Figure 2B*, *Figure 2—figure supplement 2*).

Ribosome profiling reads have a strong tendency to map with their first nucleotide occupying a particular codon position (i.e. in-frame) (*Figure 2—figure supplement 3*). We therefore examined codon position mapping across *ORF3a* to explore whether the proportions of reads in alternative frames are higher in the *ORF3c*/*ORF3d* region. Analyses were limited to reads of length 30 nt, the expected size of ribosome-bound fragments, which exhibit positioning indicative of the correct reading frame (codon position 1) of annotated genes, as well as the highest total coverage (*Figure 2—figure supplement 3*, *Figure 2—figure supplement 4*). Sliding windows reveal a peak in the fraction of reads mapping to the reading frame of *ORF3d* at its hypothesized locus, with this frame's maximum occurring at the center of *ORF3d*, co-located with a dip in the fraction of reads mapping to the frame of *ORF3a* (*Figure 2C*). These observations are reproducible across treatments (*Figure 2—figure supplement 5*), robust to sliding window size (*Figure 2—figure supplement 6*), and similar to the reading frame perturbations observed for the OLG *ORF9b* within *N* (*Figure 2—figure supplement 7*). At the same time, the *ORF3d* peak stands in contrast to the remainder of *ORF3a*, where the reading frame of *ORF3a* predominates, and smaller peaks in the frame of *ORF3d* are not accompanied by a large dip in the frame of *ORF3a* (*Figure 2—figure supplement 8*). These observations suggest that *ORF3d* is actively translated. Similar conclusions can be drawn for *ORF3c*, *ORF3d-2*, and *ORF9b*, but not *ORF9c* (*Figure 2C*, *Figure 2—figure supplement 7*, *Figure 2—figure supplement 9*).

Additional experiments have also provided evidence of ORF3d translation. *Gordon et al., 2020* used overexpression experiments to demonstrate that ORF3d (referred to as 'ORF3b'; *Table 1*) can be stably expressed and that it interacts with the mitochondrial protein STOML2. Most compellingly, ORF3d, ORF8, and N elicit the strongest antibody responses observed in COVID-19 patient sera, with ORF3d sufficient to accurately diagnose the majority of COVID-19 cases (*Hachim et al., 2020*). Because *ORF3d* is restricted to SARS-CoV-2 and pangolin-CoV (see below), this finding is unlikely to

**Table 1.** Nomenclature and reading frames for overlapping gene candidates in SARS-CoV-2 *ORF3a*.

| Gene[*] | Reading frame[†] | Genome positions, Wuhan-Hu-1 (CDS positions, *ORF3a*)[‡] | Length | Description | References |
|---|---|---|---|---|---|
| *ORF3a* | ss11 (reference) | 25393–26220 (1-828) | 276 codons (828 nt) | Ion channel formation and virus release in SARS-CoV infection; host cell apoptosis; triggers inflammation; antagonizes interferon | *Lu et al., 2010*; *Cui et al., 2019* |
| *ORF3c* | ss13 | 25457–25582 (65-190) | 42 codons (126 nt) | Features suggestive of a viroporin (*Cagliani et al., 2020*); lowest $\pi_N/\pi_S$ ratio estimated for any gene in our between-host selection analysis (*Figure 5*); overlaps codons 22–64 of *ORF3a* | First discovered by *Cagliani et al., 2020* as *ORF3h*; *ORF3c* in *Firth, 2020*; *ORF3c* in *Jungreis et al., 2020*; *ORF3a.iORF1* in *Finkel et al., 2020*; *ORF3b* in *Pavesi, 2020* |
| *ORF3d* | ss12 | 25524–25697 (132-305) | 58 codons (174 nt) | Aligned to and named *ORF3b* by *Chan et al., 2020* but is not homologous to *ORF3b*; interferon antagonism has not been demonstrated; binds STOML2 mitochondrial protein (*Gordon et al., 2020*); contains a predicted signal peptide in the region encoding *ORF3d-2*; contains an *X* motif in pangolin-CoV but not SARS-CoV-2 (*Michel et al., 2020*); may contribute to differences between SARS-CoV and SARS-CoV-2 in immune response as a unique antigenic target (*Hachim et al., 2020*; Niloufar Kavian, pers. comm.); overlaps codons 44–102 of *ORF3a* | Present study; first discovered by *Chan et al., 2020* but misclassified as *ORF3b*; *ORF3b* in *Gordon et al., 2020*, *Hachim et al., 2020*, and citing studies; 'hypothetical protein' in *Pavesi, 2020*; 'a completely different ORF' in *Michel et al., 2020* |
| *ORF3d-2* | ss12 | 25596–25697 (204-305) | 34 codons (102 nt) | A shorter isoform of *ORF3d* that starts after the first 24 codons, where the majority of premature STOP codons in SARS-CoV-2 are located; contains a predicted signal peptide (*Finkel et al., 2020*); likely expressed at higher levels than full-length *ORF3d* (*Figure 2A*) overlaps codons 68–102 of *ORF3a* | Discovered by *Finkel et al., 2020* as *ORF3a-iORF2* |
| *ORF3a-2* | ss11 (reference) | 25765–26220 (373-828) | 152 codons (456 nt) | A shorter isoform of *ORF3a* that starts after the first 124 codons; evidence of expression separate from that of *ORF3a* (*Davidson et al., 2020*); has also been conflated with *ORF3b*; equivalent to codons 124–276 of *ORF3a* | Discovered by *Davidson et al., 2020* (pers. comm.) but referred to as *ORF3b* |
| *ORF3b* region[§] | ss13 | 25814–26281; four ORFs at 25814–82, 25910–84, 26072–170, and 26183–281 (422-889; ORFs at 422–90, 518–92, 680–778, and 791–889) | 156 codons (468 nt); the four ORFs are 23, 25, 33, and 33 codons (69, 75, 99, and 99 nt) | Full-length in some related viruses, but truncated by multiple in-frame STOP codons in SARS-CoV-2; longer forms function as an interferon antagonist in SARS-related viruses; may contribute to differences between SARS-CoV and SARS-CoV-2 in immune response; although aligned to *ORF3d* by *Chan et al., 2020*, the two are not homologous; overlaps codons 141–276 of *ORF3a* (first ORF overlaps codons 141–164) | *Konno et al., 2020* claim functionality for the first (23-codon) ORF in SARS-CoV-2; the first ORF is also mentioned by *Wu et al., 2020a* |

[*]Genes are listed by position of start site in the genome from 5′ (top) to 3′ (bottom).

[†]Nomenclature as described in *Wei and Zhang, 2015* and *Nelson et al., 2020a*: ss = sense-sense (same strand); ss12 = codon position 1 of the reference frame overlaps codon position 2 of the overlapping frame on the same strand; ss13 = codon position 1 of the reference frame overlaps codon position 3 of the overlapping frame on the same strand. Frame is indicated from the perspective of *ORF3a* as the reference gene, i.e. *ORF3d* starts at codon position 3 of *ORF3a*, while *ORF3c* and *ORF3b* start at codon position 2 of *ORF3a*.

[‡]Positions and counts include STOP codons. Positions or sequences were indicated by the original publications or verified by personal communication if ambiguous.

[§]The SARS-CoV-2 region homologous to *ORF3b* of SARS-CoV contains four premature STOP codons and four distinct ORFs (AUG-to-STOP); see *Figure 4*, *Figure 6*, and *Supplementary file 1*.

be due to cross-reactivity with another coronavirus and provides strong independent evidence of ORF3d translation during infection.

## Protein sequence properties

To further investigate the antigenic properties of ORF3d, we predicted linear T cell epitopes for each SARS-CoV-2 protein. We employed NetMHCpan (*Jurtz et al., 2017*) for MHC class I (cytotoxic

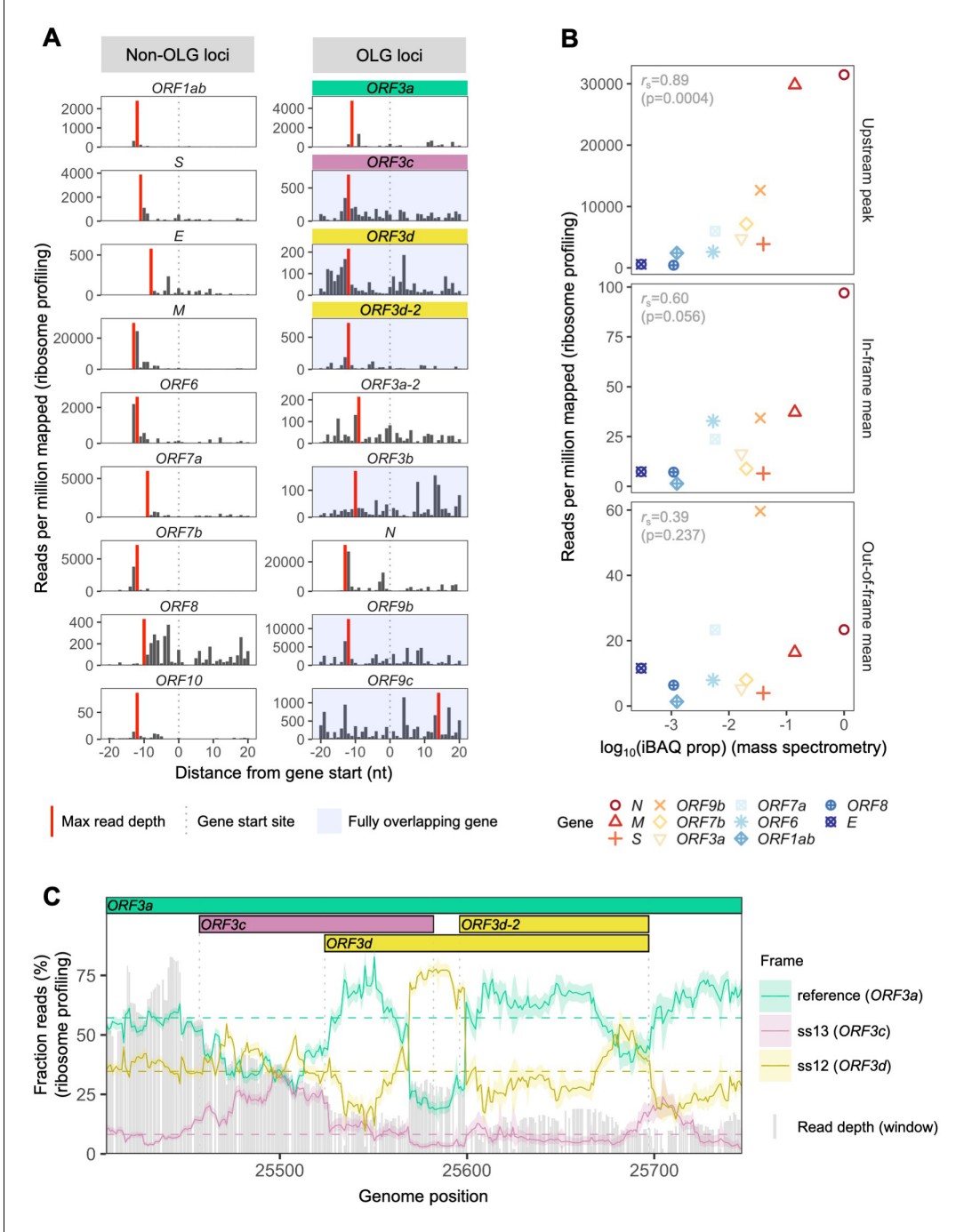

**Figure 2.** SARS-CoV-2 gene expression in ribosome profiling and mass spectrometry datasets. Ribosome profiling (Ribo-seq) data were obtained from *Finkel et al., 2020*; mass spectrometry data were obtained from *Davidson et al., 2020* and *Bezstarosti et al., 2020*. Reads were trimmed to their first (5′) nucleotide to minimize statistical dependence while preserving reading frame. Results are shown after pooling samples treated with harringtonine (SRR11713360, SRR11713361) or lactimidomycin (SRR11713358, SRR11713359) at 5 hr post-infection. (**A**) Ribosome profiling coverage (read depth) near translation initiation sites, measured as mean reads per million mapped reads. Only reads of length 29–31 nucleotides were used, chosen for their enrichment at the start sites of highly expressed annotated genes (*Figure 2—figure supplement 4*). Light blue backgrounds denote fully (internal) overlapping genes. Annotated genes show an accumulation of 5′ ends of protected reads upstream of the gene's start site (vertical gray dotted lines), peaking near the ribosome P-site offset of −12 nt (red = maximum depth). Distributions are largely consistent across individual samples (*Figure 2—figure supplement 1*). The ranges of the y-axes vary according to expression level, with the most highly expressed gene (*N*; *Figure 2—figure supplement 2*) having the largest range. (**B**) Correlation between protein expression as estimated by mass spectrometry and ribosomal profiling. 'iBAQ prop' refers to the relative (proportion of maximum) protein intensity-based absolute quantification (iBAQ) value (*Schwanhäusser et al., 2011*). Genes

*Figure 2 continued on next page*

*Figure 2 continued*

are denoted by shape and ordinally colored by iBAQ prop from high (red) to low (blue). 'Upstream peak' refers to the maximum read depth observed at the approximate P-site offset (red bars in **A**), while mean read depths were measured across each gene using 30 nt reads separately for in-frame (codon position 1) and out-of-frame (codon positions 2 and 3) sites (non-overlapping gene regions only, except for *ORF9b*). (**C**) Reading frame of ribosome profiling reads in the *ORF3c*/*ORF3d* region of *ORF3a*. Solid lines show the fraction of reads in each frame, summed across samples in sliding windows of 30 nt (step size = 1 nt; read length = 30 nt). Color denotes frame: green = reference frame (*ORF3a*); burgundy = ss13, the forward-strand frame encoding *ORF3c*, whose codon position 3 overlaps codon position 1 of *ORF3a*; and gold = ss12, the forward-strand frame encoding *ORF3d*, whose codon position 2 overlaps codon position 1 of *ORF3a*. Values are shown for the central nucleotide of each window, with shaded regions corresponding to 95% binomial confidence intervals. Alternative frame translation is suggested where a given frame (solid line) exceeds its average across the remainder of the gene (horizontal dashed line; non-OLG regions of *ORF3a*). Vertical gray dotted lines indicate gene start and end sites. Gray bars show read depth for each window, with a maximum of 2889 reads at genome position 25442.

The online version of this article includes the following source data and figure supplement(s) for figure 2:

**Source data 1.** riboseq_upstream_peaks.txt.
**Source data 2.** expression_data_by_gene_frame.txt.
**Source data 3.** riboseq_ORF3d_sliding_window.txt.
**Figure supplement 1.** Ribosome profiling coverage near translation initiation sites for individual samples.
**Figure supplement 2.** Correlation between protein expression as estimated by mass spectrometry and ribosomal profiling, for individual treatments and codon positions.
**Figure supplement 3.** Reading frames occupied by the 5′ ends of ribosome profiling reads as a function of read length.
**Figure supplement 4.** Ribosome profiling read accumulation at gene start sites as a function of read length.
**Figure supplement 5.** Reading frame of ribosome profiling reads in the *ORF3c*/*ORF3d* region of *ORF3a* for individual treatments.
**Figure supplement 6.** Reading frame of ribosome profiling reads in the *ORF3c*/*ORF3d* region of *ORF3a* for individual treatments using a smaller sliding window size of 19 nt.
**Figure supplement 7.** Reading frame of ribosome profiling reads in the *N*/*ORF9b*/*ORF9c* region.
**Figure supplement 8.** Reading frame of ribosome profiling reads in the full *ORF3a* gene.
**Figure supplement 9.** Reading frame of ribosome profiling reads in the *ORF3c*/*ORF3d* region of *ORF3a* for reads of length 29 nt.

CD8+ T cells), an approach shown to accurately predict SARS-CoV-2 epitopes shared with SARS-CoV (*Grifoni et al., 2020*), and NetMHCIIpan (*Reynisson et al., 2020*) for MHC class II (helper CD4+ T cells). Specifically, we tested all 9 amino acid (MHC I) or 15 amino acid (MHC II) substrings of each viral protein for predicted weak or strong binding by MHC. Epitope density was estimated as the mean number of predicted epitopes overlapping each residue for each protein. We also tested two sets of negative controls: (1) randomized peptides generated from each protein, representing the result expected given amino acid content alone and (2) short unannotated ORFs present in the SARS-CoV-2 genome, representing the result expected for ORFs that have been evolving without functional constraint.

For CD8+ T cells, the lowest predicted epitope density occurs in ORF3d (1.5 per residue), which is unexpected given its own amino acid content (p=0.150) and compared to short unannotated ORFs (p=0.078; permutation tests). The next lowest densities occur in N and ORF8 (*Figure 3A*). Intriguingly, as previously mentioned, these three peptides (ORF3d, N, and ORF8) also elicit the strongest antibody (B cell epitope) responses measured in COVID-19 patient sera (*Hachim et al., 2020*), suggesting a possible balance between CD8+ T and B cell epitopes. For CD4+ T cells, again, ORF3d has one of the lowest predicted epitope densities (5.6 per residue; p≥0.291), with lower values seen in only three other genes. Focusing instead on the shorter ORF3d-2 isoform, this protein contains zero predicted epitopes, which is a highly significant depletion given its own amino acid content (p=0.001) and compared to short unannotated ORFs (p=0.001). These observations suggest either a predisposition toward immune escape, allowing gene survival, or the action of selective pressures on *ORF3d* or *ORF3d-2* to remove epitopes. In stark contrast to ORF3d, ORF3c has the highest predicted CD8+ T cell epitope density (8.4 per residue), apparently as a function of its amino acid content (i.e. not differing from its randomized peptides; p=0.996). The enrichment of predicted epitopes in unannotated proteins (e.g. ORF3c and randomized peptides) but not N, for which numerous epitopes are documented (*Grifoni et al., 2020*), demonstrates that ascertainment or methodological biases cannot account for the depletion of predicted epitopes in ORF3d.

Our structural prediction for the ORF3d protein suggests α-helices connected with coils and an overall fold model that matches known protein structures (e.g. Protein Data Bank IDs 2WB7 and 6A93) with borderline confidence (*Figure 3—figure supplement 1*), similar to the predictions of

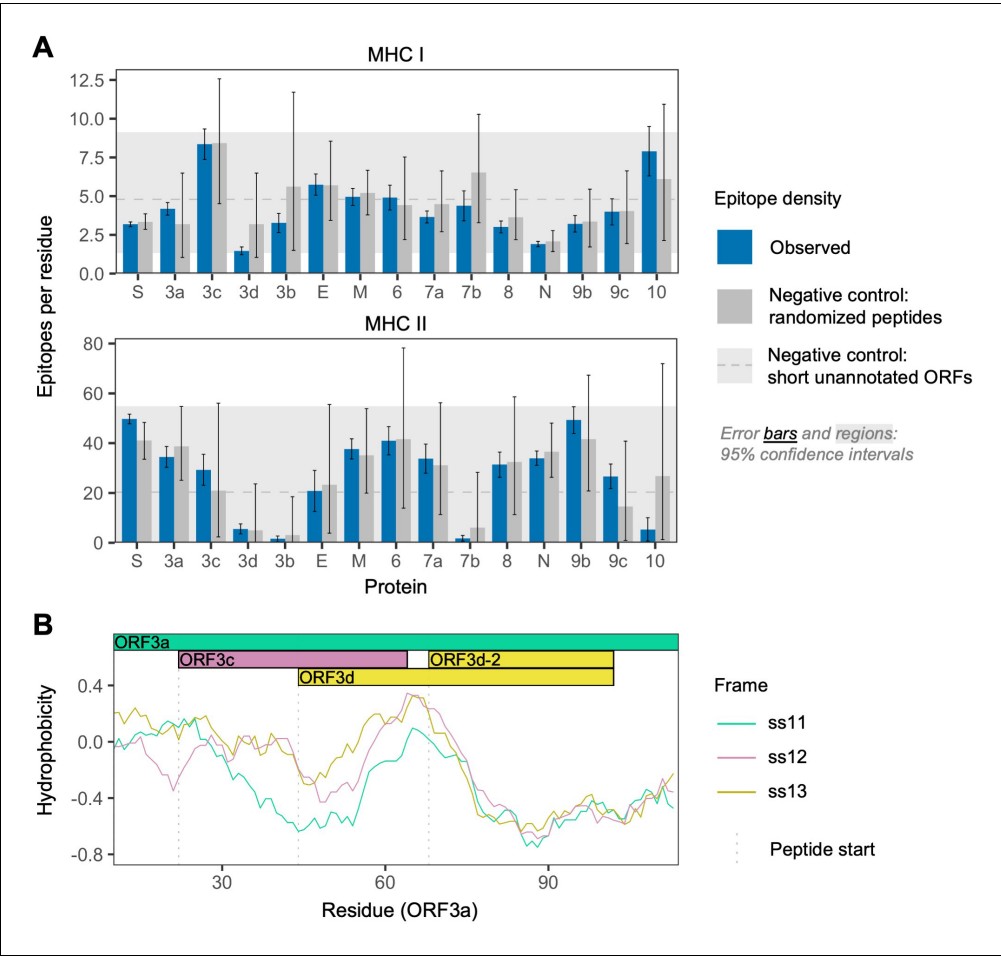

**Figure 3.** SARS-CoV-2 protein sequence properties. (**A**) Predicted densities of MHC class I-bound CD8+ T cell 9 amino acid (aa) epitopes (top) and MHC class II-bound CD4+ T cell 15 aa epitopes (bottom). Results for proteins encoded downstream of *ORF1ab* are shown. Mean numbers of predicted epitopes per residue (blue bars) are calculated as the number of epitopes overlapping each amino acid position divided by protein length. Error bars show 95% confidence intervals. Two sets of negative controls were also tested: (1) *n* = 1000 randomized peptides generated from each protein by randomly sampling its amino acids with replacement (dark gray bars), representing the result expected given amino acid content alone; and (2) short unannotated ORFs, the peptides encoded by *n* = 103 putatively nonfunctional ORFs present in the SARS-CoV-2 genome, representing the result expected for ORFs that have been evolving without functional constraint. For the short unannotated ORFs, the horizontal gray dotted line shows the mean number of epitopes per residue, and the gray-shaded region shows the 95% confidence interval (i.e. 2.5% to 97.5% quantiles). ORF3d, N, and ORF8 have the lowest MHC class I epitope densities; ORF3d, ORF3b, ORF7b, and ORF10 have the lowest MHC class II epitope densities. (**B**) Hydrophobicity profiles of amino acid sequences encoded by the three forward-strand reading frames of the *ORF3a*/*ORF3c*/*ORF3d* gene region, as calculated by the VOLPES server, using the unitless 'Factor 1' consensus hydrophobicity scale. Frame is reported using *ORF3a* as the reference, for example ss12 refers to the frame encoding *ORF3d*, for which codon position 2 overlaps codon position 1 of *ORF3a* (**Figure 2C**).

The online version of this article includes the following source data and figure supplement(s) for figure 3:

**Source data 1.** epitope_summary_MHCI_LONG.txt.
**Source data 2.** epitope_summary_MHCII_LONG.txt.
**Source data 3.** hydrophobicity_profiles_ORF3a_corr.txt.
**Figure supplement 1.** Structural prediction for the ORF3d protein.
**Figure supplement 2.** Correlations between hydrophobicity profiles of amino acid sequences encoded by the three forward-strand reading frames of the *ORF3a* region.
**Figure supplement 3.** Correlation between hydrophobicity profiles in the amino acid sequences encoded by all three forward-strand frames of *ORF3a*, by gene subregion.

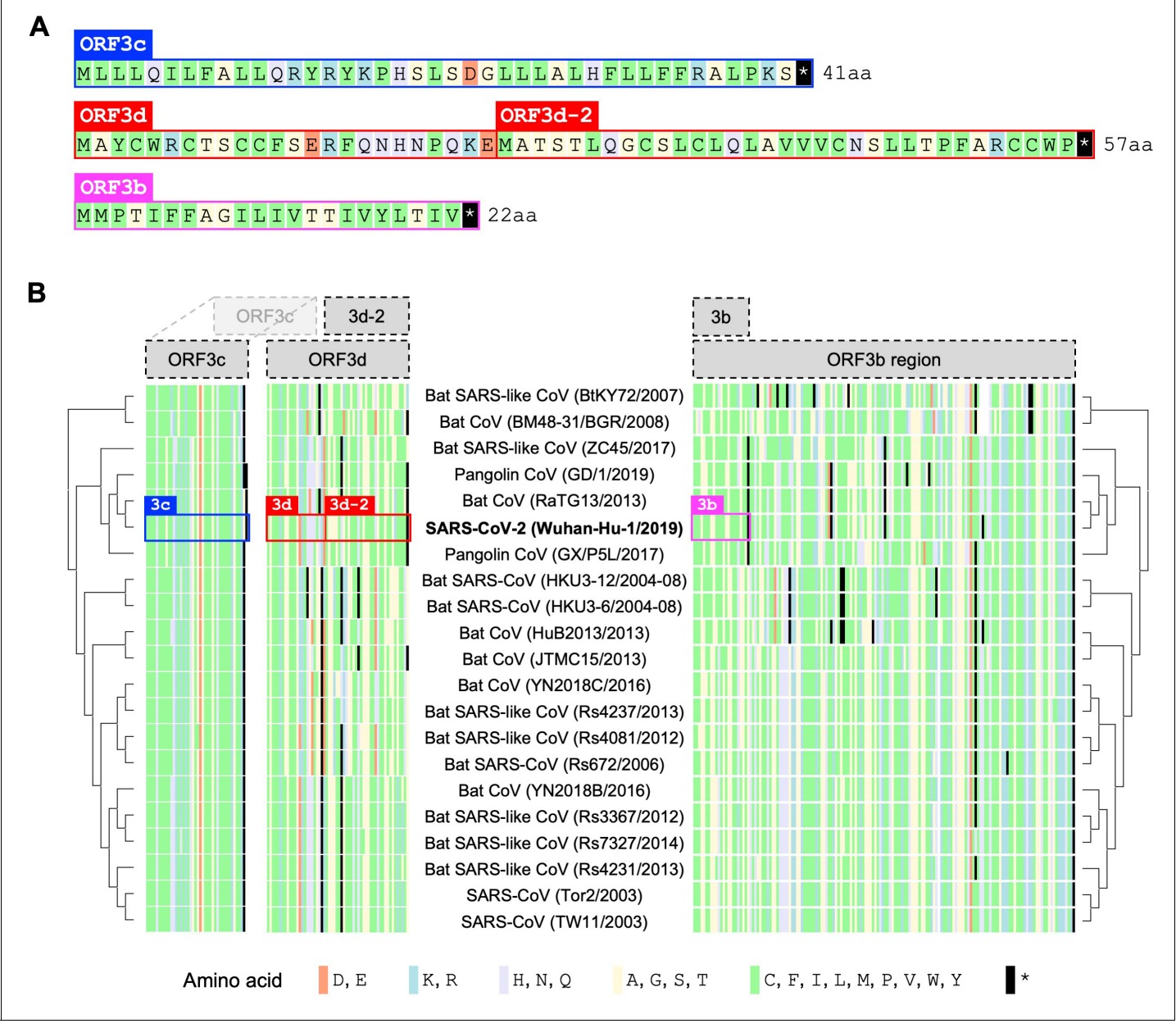

**Figure 4.** Amino acid variation in proteins encoded by genes overlapping *ORF3a* in viruses of the species *Severe acute respiratory syndrome-related coronavirus*. (**A**) Amino acid sequences of ORF3c, ORF3d, and ORF3b, as encoded by SARS-CoV-2 (Wuhan-Hu-1; NCBI: NC_045512.2). Note that, while ORF3b encodes a protein of 154 amino acids (aa) in SARS-CoV (NCBI: NC_004718.3), a premature STOP codon in SARS-CoV-2 has resulted in an ORF encoding only 22 aa. (**B**) Amino acid alignments of ORF3c, ORF3d, and ORF3b (3b) show their sequence conservation. Black lines indicate STOP codons in *ORF3d* and *ORF3b*, showing their restricted taxonomic ranges. Intact *ORF3d* is restricted to SARS-CoV-2 and pangolin-CoV GX/P5L; however, note that ORF3d-2 (denoted 3d-2), a shorter isoform of ORF3d, could have a slightly wider taxonomic range if TTG or GTG are permitted as translation initiation codons. Full-length ORF3b (ORF3b region) is found throughout members of this virus species, but truncated early in most genomes outside of SARS-CoV (*Supplementary file 1*), with the shortest isoform (denoted 3b) found in SARS-CoV-2 and closely related viruses. The online version of this article includes the following source data for figure 4:

**Source data 1.** aa_alignments_3c_3d_3b.xlsx.

*Chan et al., 2020*. Remarkably, biochemical properties that influence the structure of this novel protein appear to be inherited from the pre-existing ORF3a protein sequence encoded by the overlapping reading frame. It has recently been shown that frame-shifted nucleotide sequences tend to encode proteins with similar hydrophobicity profiles as a consequence of the standard genetic code

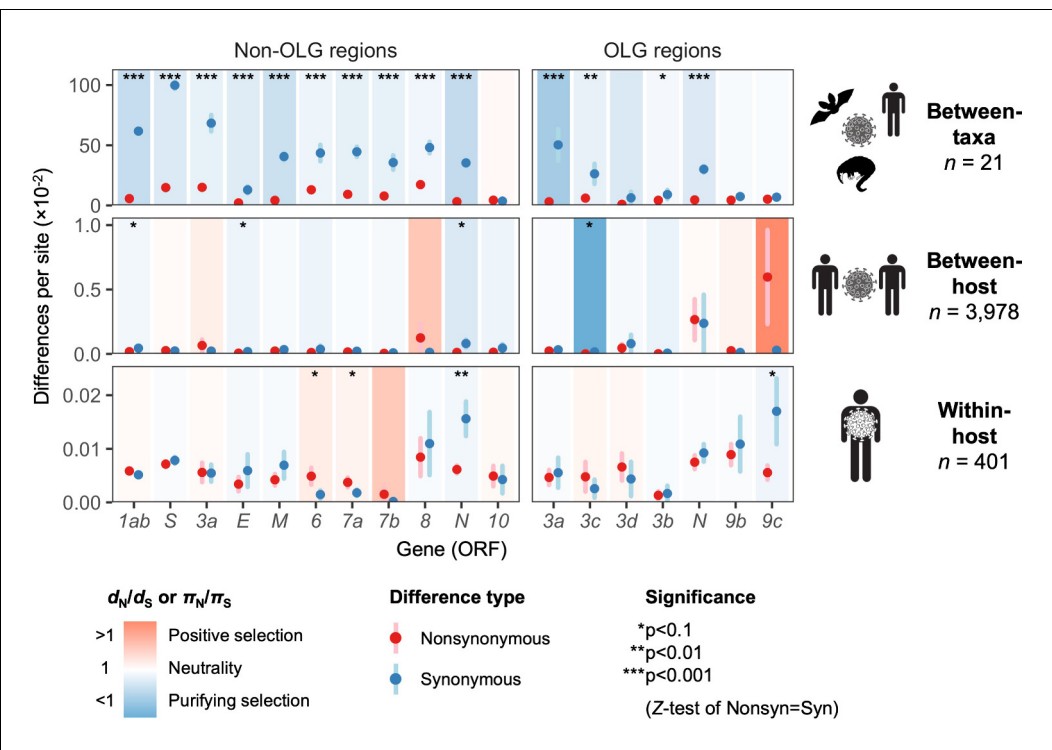

**Figure 5.** Natural selection analysis of viral nucleotide differences at three hierarchical evolutionary levels. Nucleotide differences in each virus gene were analyzed at three host levels: between-taxa divergence ($d$) among *Severe acute respiratory syndrome-related coronavirus* genomes infecting bat, human, and pangolin species; between-host diversity ($\pi$) for SARS-CoV-2 infecting human individuals (consensus-level); and within-host diversity ($\pi$) for SARS-CoV-2 infecting human individuals (deep sequencing). Each gene/level is shaded according to the ratio of mean nonsynonymous to synonymous differences per site to indicate purifying selection ($d_N/d_S < 1$ or $\pi_N/\pi_S < 1$; blue) or positive selection ($d_N/d_S > 1$ or $\pi_N/\pi_S > 1$; red). The extremely low ratio for *ORF3c* was artificially adjusted to allow the display of other ratios, and a Jukes-Cantor correction was applied to $d_N$ and $d_S$ values. Values range from a minimum of $\pi_N/\pi_S = 0.04$ (*ORF3c*, between-host; p=0.0410) to a maximum of 21.0 (*ORF9c*, between-host; p=0.126), where significance was evaluated using Z-tests of the hypothesis that $d_N-d_S = 0$ or $\pi_N-\pi_S = 0$ (10,000 bootstrap replicates, codon unit). The mean of all pairwise comparisons is shown for sequenced genomes only, i.e. no ancestral sequences were reconstructed or inferred. For each gene, sequences were only included in the between-species analysis if a complete, intact ORF (no STOPs) was present. Genes containing an overlapping gene (OLG) in a different frame were analyzed separately for non-OLG and OLG regions using SNPGenie and OLGenie, respectively. For *ORF3b*, only the region corresponding to the first ORF in SARS-CoV-2 (*Table 1*) was analyzed. The short overlap between *ORF1a* and *ORF1b* (*nsp11* and *nsp12*) was excluded from the analysis. Error bars represent the standard error of mean pairwise differences. See Materials and methods for further details.

The online version of this article includes the following source data and figure supplement(s) for figure 5:

**Source data 1.** selection_three_levels.txt.

**Figure supplement 1.** SARS-CoV-2 between-host nucleotide diversity and allele frequencies as a function of date during the initial period of the COVID-19 pandemic.

**Figure supplement 2.** Correlation between natural selection and gene expression.

---

(*Bartonek et al., 2020*). To explore whether this is the case for *ORF3d*, we used the VOLPES server (*Bartonek and Zagrovic, 2019*) to calculate the hydrophobicity profiles (*Atchley et al., 2005*) of the peptides encoded by all three frames of *ORF3a*. The maximum correlation observed occurs between the frames of *ORF3a* (ss11) and *ORF3d* (ss12) in the region encoding ORF3d (ORF3a residues 64–102), with $r_S = 0.87$ (p=$5.79 \times 10^{-13}$, Spearman's rank), which is much stronger than the correlation between these frames observed for the non-OLG residues of ORF3a ($r_S = 0.27$, p=$8.70 \times 10^{-4}$) (*Figure 3C*; *Figure 3—figure supplement 2*). This conservation of a structure-related property with the ORF3a protein provides further evidence for *ORF3d* functionality, and may predispose this

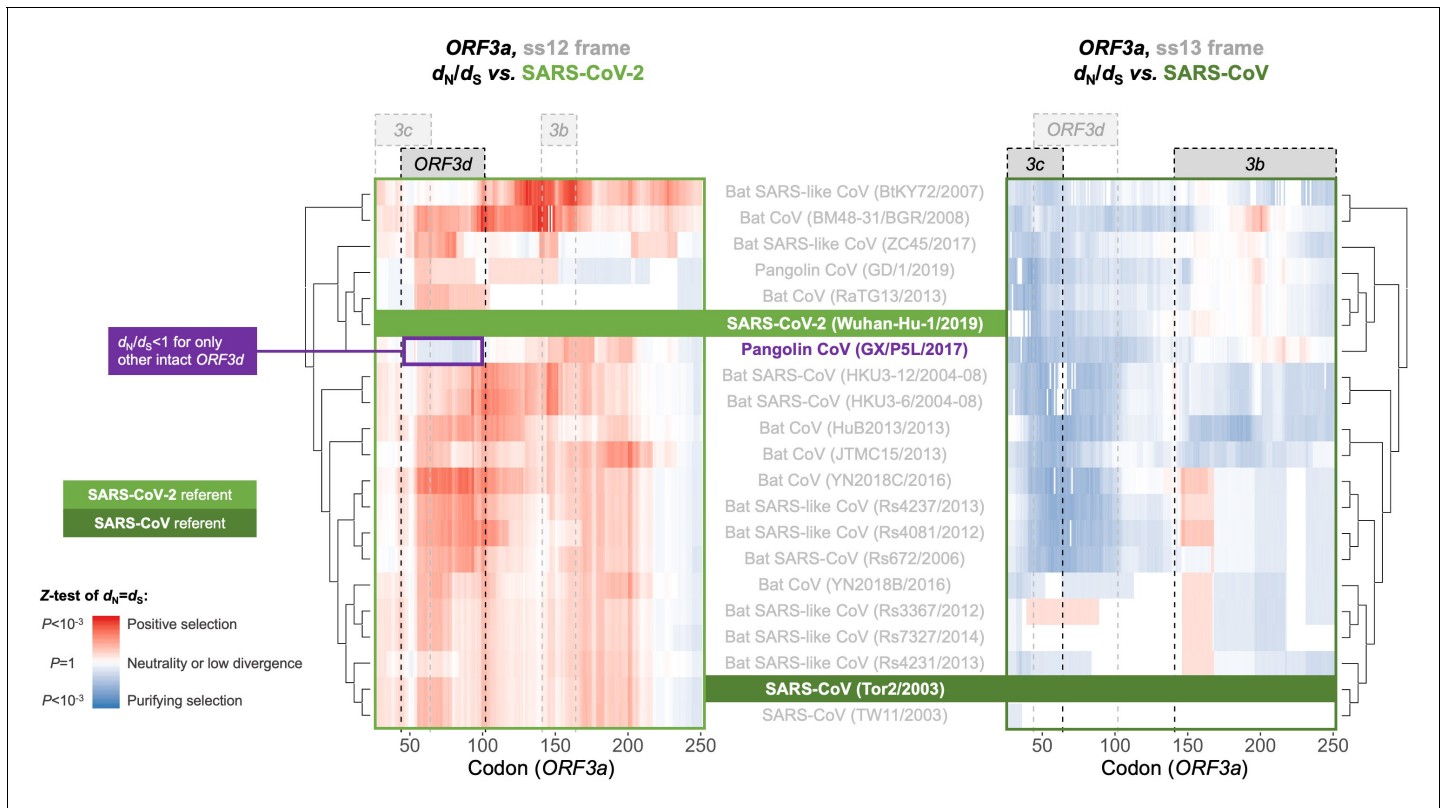

**Figure 6.** Between-taxa sliding window analysis of natural selection on overlapping frames of *ORF3a*. Pairwise sliding window analysis (window size = 50 codons; step size = 1 codon) of selection across members of the species *Severe acute respiratory syndrome-related coronavirus*. OLG-appropriate $d_N/d_S$ values were computed using OLGenie (**Nelson et al., 2020a**), a method that is conservative (non-conservative) for detecting purifying (positive) selection, and a Jukes-Cantor correction for multiple hits was employed. On the left-hand side, each genome is compared to SARS-CoV-2 in the ss12 reading frame of *ORF3a*, which contains *ORF3d* (**Table 1** and **Figure 2C**). This frame shows evidence for purifying selection specific to the *ORF3d* region that is limited to the comparison with pangolin-CoV GX/P5L. On the right-hand side, this analysis is repeated for the ss13 reading frame of *ORF3a*, which contains *ORF3c* and *ORF3b* (**Table 1** and **Figure 2C**), this time with respect to SARS-CoV, where full-length *ORF3b* is functional. This frame shows constraint across much of this gene and virus species, and *ORF3c* in particular is deeply conserved.

The online version of this article includes the following source data and figure supplement(s) for figure 6:

**Source data 1.** SARS-CoV-2-ref_ORF3a_ss12_windows.txt.
**Source data 2.** SARS-CoV-ref_ORF3a_ss13_windows.txt.
**Figure supplement 1.** Between-taxa sliding window of genes overlapping *N*.

region toward de novo gene birth. Again, *ORF3c* shows the opposite pattern, as the minimum correlation observed between hydrophobicity profiles occurs between the frames of *ORF3a* (ss11) and *ORF3c* (ss13) in the region encoding ORF3c (ORF3a residues 22–43), with $r_S = -0.40$ (p=6.16 $\times$ 10$^{-2}$). Such disparate hydrophobicity profiles may be due to the strong conservation of *ORF3c* (see below).

## *ORF3d* taxonomic distribution and origins

To assess the origin of *ORF3d* and its conservation within and among host taxa, we aligned 21 *Severe acute respiratory syndrome-related coronavirus* genomes reported by **Lam et al., 2020** (**Supplementary file 1**), limiting our analysis to those with an annotated *ORF1ab* and no frameshift variants relative to SARS-CoV-2 in the core genes *ORF1ab*, *S*, *ORF3a*, *E*, *M*, *ORF7a*, *ORF7b*, and *N* (**Supplementary file 1**; SARS-related-CoV_ALN.fasta, supplementary data). All other genes are also intact (i.e. there are no mid-sequence STOP codons) in all genomes (**Supplementary file 1**), with the exception of *ORF3d*, *ORF3b*, and *ORF8* (**Figure 4**). Specifically, *ORF3d* is intact in only two sequences: SARS-CoV-2 Wuhan-Hu-1 and pangolin-CoVs from Guangxi (GX/P5L). Full-length *ORF3b* is intact in only three sequences: SARS-CoV TW11, SARS-CoV Tor2, and bat-CoV Rs7327, with the

remainder having premature STOP codons (*Supplementary file 1*). Finally, *ORF8* is intact in all but five sequences, where it contains premature STOP codons or large-scale deletions (*Figure 1B*).

The presence of intact *ORF3d* homologs among viruses infecting different host species (human and pangolin) raises the possibility of functional conservation. However, the taxonomic distribution of this ORF is incongruent with whole-genome phylogenies in that *ORF3d* is intact in the pangolin-CoV more distantly related to SARS-CoV-2 (GX/P5L) but not the more closely related one (GD/1) (*Figure 4*), a finding confirmed by the alignment of *Boni et al., 2020*. New sequence data reveal similarly puzzling trends: *ORF3d* contains STOP codons in the closely related bat-CoV RmYN02 (GISAID: EPI_ISL_412977; data not shown), but it is intact in three more distantly related bat-CoVs discovered in Rwanda and Uganda, where it is further extended by 20 codons (total 78 codons) but shows no evidence of conservation with SARS-CoV-2 (*Wells et al., 2020* and pers. comm; data not shown). Further, phylogenies of the 21 *Severe acute respiratory syndrome-related coronavirus* genomes built on *ORF3a* are incongruent with whole-genome phylogenies, likely due to the presence of recombination breakpoints in *ORF3a* near *ORF3d* (*Boni et al., 2020*; *Rehman et al., 2020*). Recombination, convergence, or recurrent loss may therefore have played a role in the origin and taxonomic distribution of *ORF3d*.

## Between-taxa divergence

To estimate natural selection on *ORF3d*, we measured viral diversity at three hierarchical evolutionary levels: between-taxa, between-host, and within-host. Specifically, between-taxa refers to divergence ($d$) among the 21 aforementioned viruses infecting bat, human, or pangolin (*Figure 1B*); between-host refers to diversity ($\pi$) between consensus-level SARS-CoV-2 genomes infecting different human individuals; and within-host refers to $\pi$ in deeply sequenced 'intrahost' SARS-CoV-2 samples from single human individuals. At each level, we inferred selection by estimating mean pairwise nonsynonymous ($d_N$ or $\pi_N$; amino acid changing) and synonymous ($d_S$ or $\pi_S$; not amino acid changing) distances among all sequences. Importantly, we combined standard (non-OLG) methods (*Nei and Gojobori, 1986*; *Nelson et al., 2015*) with OLGenie, a new $d_N/d_S$ method tailored for OLGs (hereafter OLG $d_N/d_S$), which we previously used to verify purifying selection on a novel OLG in HIV-1 (*Nelson et al., 2020a*).

The only gene to show significant evidence of purifying selection at all three evolutionary levels is the nucleocapsid gene *N* (*Figure 5*), which undergoes disproportionately low rates of nonsynonymous change ($d_N/d_S < 1$ and $\pi_N/\pi_S < 1$) specifically in its non-OLG regions, evidencing strict functional constraint. *N* is also the most highly expressed gene (*Figure 2—figure supplement 2*), confirming that selection has more opportunity to act when a protein is manufactured in abundance (*Figure 2B*; *Supplementary file 1*) (Materials and methods). Importantly, this signal can be missed if non-OLG methods are applied to *N* without accounting for its internal OLGs, *ORF9b* and *ORF9c* (e.g. at the between-host level, p=0.0268 when excluding OLG regions, but p=0.411 when including them; *Supplementary file 1*).

With respect to *ORF3d*, comparison of Wuhan-Hu-1 to pangolin-CoV GX/P5L (NCBI: MT040335.1) yields OLG $d_N/d_S$ = 0.14 (p=0.264) (*Figure 5*), whereas inclusion of a third allele found in pangolin-CoV GX/P4L (NCBI: MT040333.1) yields 0.43 (p=0.488) (*Supplementary file 1*). Because this is suggestive of constraint, we performed sliding windows of OLG $d_N/d_S$ across the length of *ORF3a*. Pairwise comparisons of each sequence to SARS-CoV-2 reveal OLG $d_N/d_S < 1$ that is specific to the reading frame, genome positions, and species in which *ORF3d* is intact (pangolin-CoV GX/P5L) (*Figure 6*, left). This signal is independent of whether STOP codons are present, so its consilience with the only intact ORF in this region and species is highly suggestive of purifying selection. We note that this conclusion does not contradict studies which fail to find evidence of *ORF3d* conservation when comparing taxa where *ORF3d* is absent (e.g. *Jungreis et al., 2020*), because *ORF3d* is a novel gene and would by definition lack such conservation. This contrastive signal is also similar to what is observed for the known OLGs *ORF3b* (in comparisons to SARS-CoV; *Figure 6*, right), and *ORF9b* and *ORF9c* (in both viruses; *Figure 6—figure supplement 1*).

## Between-host evolution and pandemic spread

Purifying selection can be specific to just one taxon, as in the case of novel genes. Thus, to measure selection within SARS-CoV-2 only, we obtained 3978 high-quality human SARS-CoV-2 consensus

**Table 2.** The mutational path to European pandemic founder haplotypes[*].

| Variant | EP–3 | EP–2 | EP[†] | EP+1[†] | EP+1+LOF |
|---|---|---|---|---|---|
| C241U (5′-UTR) | - | - | + | + | + |
| C3037U (nsp3-F106F) | - | - | + | + | + |
| C14408U (RdRp-P323L) | - | - | - | + | + |
| A23403G (Spike-D614G) | - | + | + | + | + |
| G25563U (ORF3a-Q57H/ORF3c-R36I/ORF3d-E14*) | - | - | - | - | + |
| | | | | | |
| Earliest collection[§] | 24-Dec | 7-Feb | 28-Jan | 20-Feb | 21-Feb |
| Earliest location[§] | Wuhan | Wuhan | Munich (Shanghai)[‡] | Lombardy | Hauts de France |
| Occurrence in China | 233 | 1 | 1 (2)[‡] | 0 | 0 |
| Occurrence in Europe | 458 | 0 | 21 | 1153 | 310 |
| Occurrence in Italy | 1 | 0 | 0 | 27 | 0 |
| Occurrence in Germany | 15 | 0 | 1 | 11 | 21 |
| Occurrence in Belgium | 27 | 1 | 20 | 187 | 27 |
| Occurrence in UK | 210 | 0 | 0 | 338 | 38 |
| Occurrence in Iceland | 56 | 0 | 0 | 212 | 54 |
| Occurrence in France | 14 | | 0 | 72 | 102 |
| Occurrence in US | 467 | 0 | 0 | 88[**] | 326[**] |
| Total in GISAID[††] | 1610 | 2 | 22 | 1455 | 752 |

[*]Haplotypes are here defined by the presence (+) or absence (-) of five high-frequency variants (rows 1–5), and other variants with lower frequencies on these backgrounds are ignored. EP-1 is not observed in our dataset.

[†]The EP haplotype is first detected in German patient #4 and is a documented founder for coronavirus spread in Germany (***Rothe et al., 2020***). Neither the EP nor EP+1 haplotypes were detectable between January 28 and February 20, although they immediately became a major haplotype once EP+1 was detectable. Failure to detect these two haplotypes during these 3 weeks could potentially be explained by ascertainment bias, for example lack of testing for travel-independent cases.

[‡]This Shanghai sample (GISAID: EPI_ISL_416327) comprises 1.32% poly-Ns and failed our quality control criteria, but is added here since it is potentially relevant to the origin of the EP haplotype. Including this sample, the EP haplotype is observed in Shanghai twice.

[§]The earliest collection location and time are highly subject to collection and submission bias and do not necessarily reflect where the mutation/haplotype first occurred.

[**]There is likely a testing bias in the United States, as the EP+1+LOF haplotype was often detected in Washington but EP+1 was not.

[††]These numbers are based on 3853 samples from December 24 to April 1 at the time of GISAID accession that passed both our quality control criteria for alignment and for this particular analysis (i.e. no ambiguous genotype calls among the five SNPs in this table), unless otherwise stated.

sequences from GISAID (accessed April 10, 2020; *Supplementary file 1*; *Supplementary file 2*). Between-host diversity was sufficient to detect marginally significant purifying selection across all genes ($\pi_N/\pi_S$ = 0.50, p=0.0613, Z-test; *Figure 5—figure supplement 1*) but not most individual genes (*Figure 5*). Therefore, we instead investigated single mutations over time, limiting to 27 high-frequency variants (minor allele frequency ≥2%; *Supplementary file 1*).

One high-frequency mutation occurred in *ORF3d*: G25563U, here denoted *ORF3d*-LOF (*ORF3d*-loss-of-function). This mutation causes a STOP codon in *ORF3d* (ORF3d-E14*) but nonsynonymous changes in both *ORF3a* (ORF3a-Q57H) and *ORF3c* (ORF3c-R36I), another OLG that overlaps this site (*Figure 1A*). *ORF3d*-LOF is not observed in any other species member included in our analysis (*Figure 4*; SARS-related-CoV_ALN.fasta, supplementary data). During the first months of the COVID-19 pandemic, *ORF3d*-LOF increased in frequency (*Supplementary file 1*) in multiple locations (*Figure 5—figure supplement 1D*; *Supplementary file 1*), making this mutation a candidate for natural selection on ORF3a, ORF3c, ORF3d, or any combination thereof (*Kosakovsky-Pond, 2020*). However, temporal allele frequency trajectories (*Figure 5—figure supplement 1D*) and similar signals from phylogenetic branch tests may also be caused by founder effects or genetic drift, and are

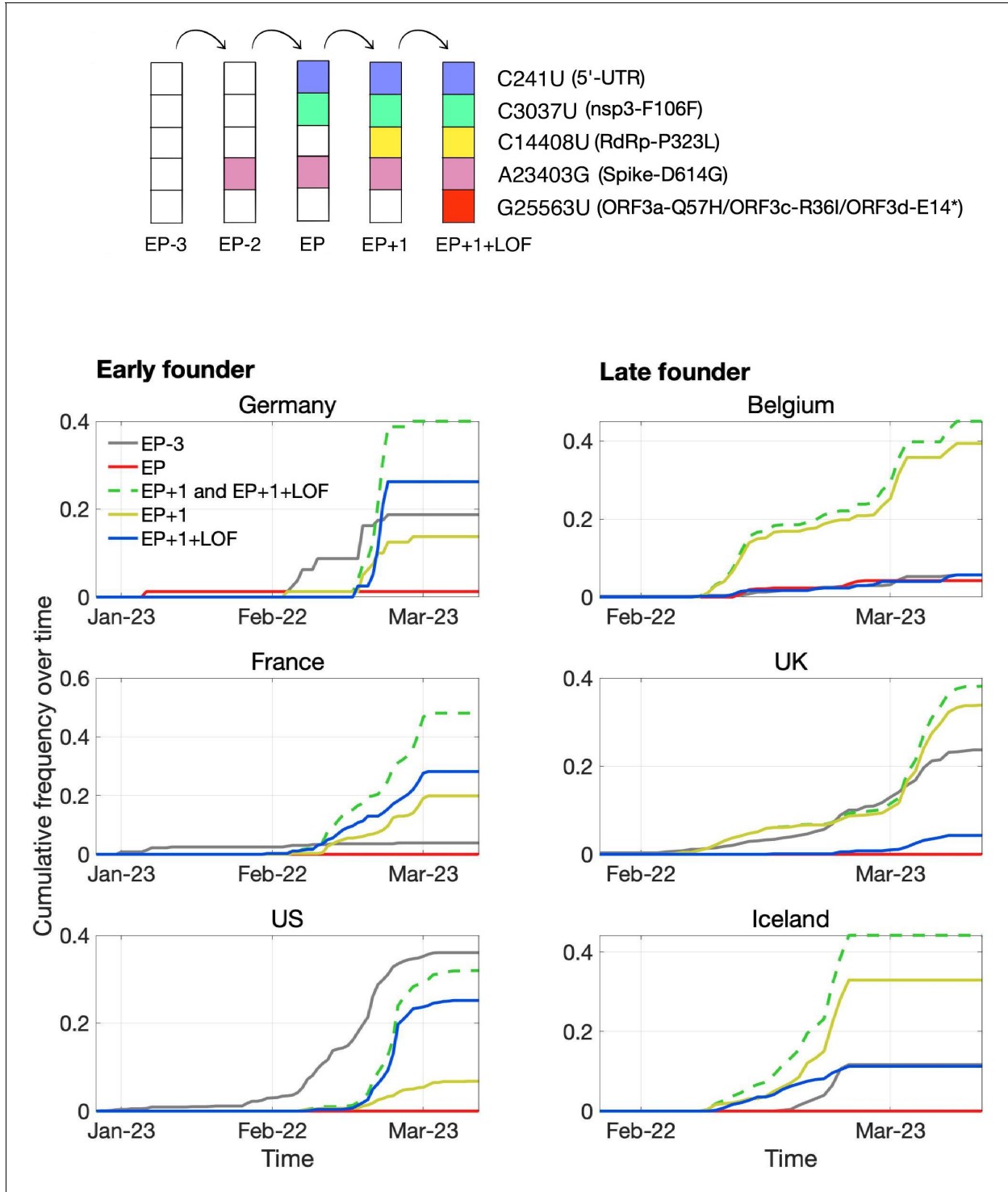

**Figure 7.** Pandemic spread of the EP+1 haplotype and the hitchhiking of *ORF3d*-LOF. The mutational path leading to EP+1+LOF is shown in the upper panel. Cumulative frequencies of haplotypes in samples from Germany and five other countries with the most abundant sequence data are shown in the lower panel. Countries are grouped into early founder (left) and late founder (right) based on the presence or absence of SARS-CoV-2 samples from January, respectively. In the early founder group, EP–3 (gray) is observed earlier than other haplotypes in France and the US, and EP (red) is observed early in Germany, giving them the advantage of a founder effect. However, neither EP nor EP–3 dominate later spread; instead, EP+1 (yellow) and EP+1+LOF (blue) increase much faster despite their later appearance in these countries. In the late founder group, multiple haplotypes appear at approximately the same time, but EP–3 and EP spread more slowly. The green dashed line shows the combined frequencies of EP+1 and EP+1+LOF (yellow and blue, respectively). Note that EP–1 is never observed in our dataset.

susceptible to ascertainment bias (e.g. preferential sequencing of imported infections and uneven geographic sampling) and stochastic error (e.g. small sample sizes).

To partially account for these confounding factors, we instead constructed the mutational path leading from the SARS-CoV-2 haplotype collected in December 2019 to the haplotype carrying *ORF3d*-LOF. This path involves five mutations (C241U, C3037U, C14408U, A23403G, G25563U), constituting five observed haplotypes (EP–3 → EP–2 → EP → EP+1 → EP+1+LOF), shown in *Table 2*. Here, EP is suggested to have driven the European Pandemic (detected in German patient #4; see footnote 3 of *Table 2*; *Korber et al., 2020*; *Rothe et al., 2020*); EP–3 is the Wuhan founder haplotype; EP–1 is never observed in our dataset; and LOF refers to *ORF3d*-LOF. We then documented the frequencies and earliest collection date of each haplotype (*Table 2*) to determine when *ORF3d*-LOF occurred on the EP background.

Surprisingly, despite its expected predominance in Europe due to a founder effect, the EP haplotype is extremely rare. By contrast, haplotypes with one additional mutation (C14408U; RdRp-P323L) on the EP background are common in Europe, and *ORF3d*-LOF occurred very early on this background to create EP+1+LOF from EP+1. Neither of these two haplotypes was initially observed in China (*Table 2*), suggesting that they might have arisen in Europe. Thus, we further partitioned the samples into two groups, corresponding to countries with early founders (January samples) or only late founders (no January samples) (*Figure 7*). In the early founder group, EP–3 (Wuhan) is the first haplotype detected in most countries, consistent with most early COVID-19 cases being related to travel from Wuhan. Because this implies that genotypes EP–3 and EP had longer to spread in the early founder group, it is surprising that their spread is dwarfed by an increase in EP+1 and EP+1+LOF starting in late February. This turnover is most evident in the late founder group, where multiple haplotypes are detected in a narrow time window, and the number of cumulative samples is always dominated by EP+1 and EP+1+LOF. Thus, while founder effects and drift are plausible explanations, it is also worth investigating whether the early spread of *ORF3d*-LOF may have been caused by its linkage with another driver, either C14408U (+1 variant) or a subsequent variant(s) occurring on the EP+1+LOF background (Discussion).

## Within-host diversity and mutational bias

Examination of within-host variation across multiple samples allows detection of recurrent mutations, which might indicate mutation bias or within-host selection. To investigate these possibilities, we obtained 401 high-depth 'intrahost' human SARS-CoV-2 samples from the Sequence Read Archive (*Supplementary file 1*) and called SNPs relative to the Wuhan-Hu-1 reference genome. Within human hosts, 42% of SNPs passed our false-discovery rate criterion (Materials and methods), with a median passing minor allele frequency of 2% (21 of 1344 reads). Using these variants to estimate within-host diversity, *ORF3d* does not show significant evidence of selection, with OLG $\pi_N/\pi_S$ = 1.5 (p=0.584) (*Figure 5*). We also examined six high-depth samples of pangolin-CoVs from Guangxi, but no conclusions could be drawn due to low sequence quality (Materials and methods; *Supplementary file 1*).

To identify recurrent mutations in multiple SARS-CoV-2 samples, separately for each site, we limited to samples for which the major or fixed allele is also ancestral (i.e. matches Wuhan-Hu-1). At such sites, precluding sequencing artifacts or coinfection by multiple genotypes, minor alleles occurring in more than one sample are expected to be derived and recurrent (i.e. identical by state but not descent). For *ORF3d*, we observe *ORF3d*-LOF as a minor allele in three (0.94%) of 320 samples (frequencies of 20.7%, 6.0%, and 2.2% in samples SRR11410536, SRR11479046, and SRR11494643, respectively). This proportion of samples with a recurrent mutation is high but not unusual, as 1.7% of observed minor variants have an equal or higher proportion of recurrence. Additionally, no mutations in *ORF3d* recur in >2.5% of samples (*Figure 8*). Thus, we find no evidence that within-host selection or mutation pressure are involved in the spread of *ORF3d*-LOF. However, a small number of other loci exhibit high rates of recurrent mutation, with five mutations independently observed in ~10% of samples or more (Materials and methods; *Figure 8*). Surprisingly, another STOP mutation (A404U; NSP1-L47*) is never a major allele but is observed at low frequencies in 44% of samples (*Figure 8*, *Figure 8—figure supplement 1*), unexplainable by mutational bias and warranting investigation.

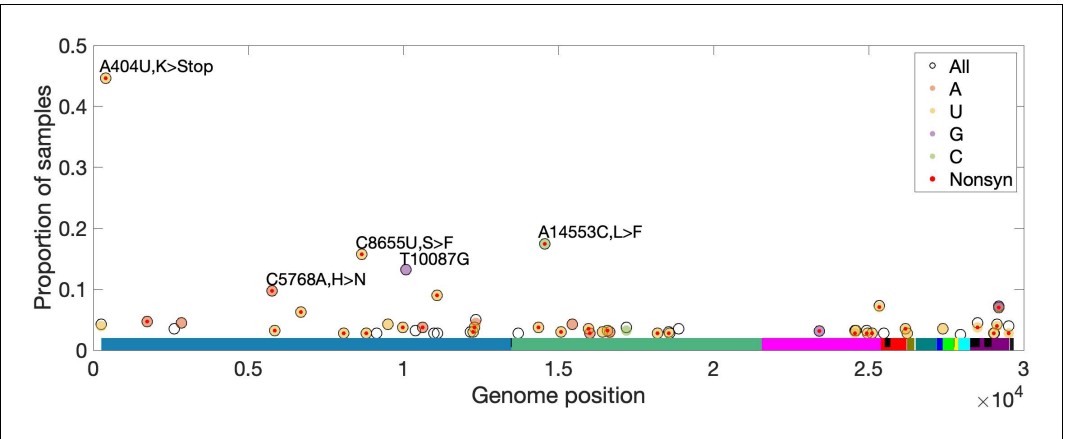

**Figure 8.** High-frequency within-host mutations. For each site, mutations that occur in more than 2.5% of samples are shown, limiting to samples where the major or fixed allele matches Wuhan-Hu-1 at that site. The y-axis shows the proportion of such samples having the indicated minor (derived) allele. Each locus has up to three possible single-nucleotide derived alleles compared to the reference background. Open circles (black outlines) show the proportion of samples having any of the three possible derived alleles ('All'), while solid circles (color fill) show the proportion of samples having a specific derived allele (equivalent to 'All' if only one variant is observed). For most sites, only one derived mutation type (e.g. C→U) is observed across all samples. Precluding co-infection by multiple genotypes and sequencing errors, derived mutations occurring in more than one sample (y-axis) must be identical by state but not descent (i.e. recurrent). Genome positions are plotted on the x-axis, with distinct genes shown in different colors and overlapping genes shown as black blocks within reference genes. Nonsynonymous and nonsense mutations (together denoted 'Nonsyn') are indicated with a red dot. Source data are available in the supplementary material.

The online version of this article includes the following figure supplement(s) for figure 8:

**Figure supplement 1.** Recurrent nonsynonymous mutations observed in multiple human hosts.

## Discussion

Our analyses provide strong evidence that SARS-CoV-2 contains a novel overlapping gene (OLG), *ORF3d*, that has not been consistently identified or fully analyzed before this study. The annotation of a newly emerged virus is difficult, particularly in genome regions subject to frequent gene gain and loss. Moreover, due to the inherent difficulty of studying OLGs, they tend to be less carefully documented than non-OLGs; for example, *ORF9b* and *ORF9c* are still not annotated in the most-used reference genome, Wuhan-Hu-1 (last accessed September 26, 2020). Therefore, de novo and homology-based annotation are both essential, followed by careful expression analyses using multi-omic data and evolutionary analyses within and between species. In particular, we emphasize the importance of using whole-gene or genome alignments when inferring homology for both OLGs and non-OLGs, taking into account genome positions and all reading frames. Unfortunately, in the case of SARS-CoV-2, the lack of such inspections has led to mis-annotation and a domino effect. For example, homology between *ORF3b* (SARS-CoV) and *ORF3d* (SARS-CoV-2) has erroneously been implied, leading to unwarranted inferences of shared functionality (*Table 1*). Given the rapid growth of the SARS-CoV-2 literature, it is likely this mistake will be further propagated. We therefore provide a detailed annotation of Wuhan-Hu-1 protein-coding genes and codons in *Supplementary file 1*, as a resource for future studies.

Our study highlights the highly dynamic process of frequent losses and gains of accessory genes in the species *Severe acute respiratory syndrome-related coronavirus*, with the greatest functional constraint typically observed for the most highly expressed genes (*Figure 5—figure supplement 2*). With respect to gene loss, while many accessory genes may be dispensable for viruses in cell culture, they often play an important role in natural hosts (*Forni et al., 2017*). Thus, their loss may represent a key step in adaptation to new hosts after crossing a species barrier (*Gorbalenya et al., 2006*). For example, the absence of full-length *ORF3b* in SARS-CoV-2 has received attention from only a few authors (e.g. *Lokugamage et al., 2020*), even though it plays a central role in SARS-CoV infection and early immune interactions as an interferon antagonist (*Kopecky-Bromberg et al., 2007*), with

effects modulated by ORF length (*Zhou et al., 2012*). Thus, the absence or truncation of *ORF3b* in SARS-CoV-2 may be immunologically important (*Yuen et al., 2020*), for example in the suppression of type I interferon induction (*Konno et al., 2020*; *Lokugamage et al., 2020*). Further, loss of function (LOF) mutations in *ORF3d* and any other ORFs need to be taken into account for testing or therapeutics, particularly in light of predicted cellular interactions (*Gordon et al., 2020*).

With respect to gene gain, the apparent presence of *ORF3d* coincident with the inferred entry of SARS-CoV-2 into humans from a hitherto undetermined reservoir host suggests that this gene is functionally relevant for the emergent properties of SARS-CoV-2, analogous to *asp* for HIV-1-M (*Cassan et al., 2016*). The mechanisms and detailed histories of gene birth, neo-functionalization, and survival of novel genes remain unclear. However, our findings that ORF3d remarkably inherited its hydrophobicity profile from the overlapping region of ORF3a, that ORF3d is depleted for predicted CD8+ and CD4+ T cell epitopes, and that *ORF3d* exhibits potential purifying selection between SARS-CoV-2 and pangolin-CoV GX/P5L, provide strong leads for further research. Indeed, both the structural and immunogenic properties of new viral proteins deserve further attention and are important parts of their evolutionary story.

Failure to account for OLGs can lead to erroneous inference of (or failure to detect) natural selection in known genes, and vice versa. For example, a synonymous variant in one reading frame is very likely to be nonsynonymous in a second overlapping frame. As a result, purifying selection in the second frame usually lowers $d_S$ (raises $d_N/d_S$) in the first frame, increasing the likelihood of mis-inferring positive selection (*Holmes et al., 2006*; *Sabath et al., 2008*; *Nelson et al., 2020a*). Such errors could, in turn, lead to mischaracterization of the genetic contributions of OLG loci to important viral properties such as incidence and persistence. One potential consequence is misguided countermeasure efforts, for example through failure to detect functionally conserved or immunologically important genome regions.

Although our study focuses on *ORF3d*, other OLGs have been proposed in SARS-CoV-2 and warrant investigation. *ORF3d-2*, a shorter isoform of *ORF3d* (*Table 1*), shows evidence of higher overall expression than *ORF3d* (*Figure 2A*). However, analysis of full-length *ORF3d* is confounded by its 20 codon triple overlap with both *ORF3a* and *ORF3c*, making expression and selection results difficult to compare between the two isoforms. Although not evolutionarily novel, the recently discovered *ORF3c* (*Table 1*) shows deep conservation among viruses of this species (*Cagliani et al., 2020*; *Firth, 2020*; *Jungreis et al., 2020*), the lowest $\pi_N/\pi_S$ ratio observed in our between-host analysis (*Figure 5*), strong evidence for translation in SARS-CoV-2 (*Figure 2*), and the highest predicted CD8+ T cell epitope density (*Figure 3*). Within *S*, *S-iORF2* (genome positions 21768–21863) also shows evidence of translation from ribosome profiling (*Finkel et al., 2020*), and between-host comparisons suggest purifying selection ($\pi_N/\pi_S$ = 0.22, p=0.0278) (*Table 1*). We therefore suggest that the SARS-CoV-2 genome could contain additional undocumented or novel OLGs.

Our comprehensive evolutionary analysis of the SARS-CoV-2 genome demonstrates that many genes are under relaxed purifying selection, consistent with the exponential growth of the virus (*Gazave et al., 2013*). At the between-host level, nucleotide diversity increases somewhat over the initial period of the COVID-19 pandemic, tracking the number of locations sampled, while the $\pi_N/\pi_S$ ratio remains relatively constant at 0.46 (±0.030 SEM) (*Figure 5—figure supplement 1B*). Other genes differ in the strength and direction of selection at the three evolutionary levels, for example *ORF9c* (*Figure 5*), suggesting a shift in function or importance over time or between different host species or individuals. *ORF3d* and *ORF8* are among the youngest genes in SARS-CoV-2, being taxonomically restricted to a subset of betacoronaviruses (*Cui et al., 2019*), and both exhibit high rates of change and turnover (*Figure 1*; *Figure 5*; SARS-related-CoV_ALN.fasta, supplementary data). High between-host $\pi_N/\pi_S$ was also observed in *ORF8* of SARS-CoV, perhaps due to a relaxation of purifying selection upon entry into civet cats or humans (*Forni et al., 2017*). However, *ORF3d* and *ORF8* both exhibit strong antibody (B cell epitope) responses (*Hachim et al., 2020*) and predicted T cell epitope depletion (*Figure 3*) in SARS-CoV-2. This highlights the important connection between evolutionary and immunologic processes (*Daugherty and Malik, 2012*), as antigenic peptides may impose a fitness cost for the virus by allowing immune detection. Thus, the loss or truncation of these genes may share an immunological basis and deserves further attention.

The quick expansion of *ORF3d*-LOF (EP+1+LOF) and its background (EP+1) during this pandemic is surprising, given that a founder effect would have favored other variants that arrived earlier. However, newer data show that *ORF3d*-LOF has remained at relatively low frequencies compared to EP

+1 variants (*Kosakovsky-Pond, 2020*), and plausible explanations exist which do not require invoking natural selection. First, the slower 'spread' of EP–3 and EP may be an artifact of a sampling policy bias in case isolation: testing and quarantine were preferentially applied to travellers who had recently visited Wuhan, which may have led to selective detection, isolation, and tracing of these haplotypes rather than those with mutations occurring within Europe (e.g. C14408U and G25563U). (However, the EP+1 and EP+1+LOF haplotypes also increase faster in the late founder group, where it is unclear which haplotype was more travel-related.) Second, it is possible that the EP haplotype was effectively controlled, while EP+1 was introduced independently (*Worobey et al., 2020*). Third, although G25563U itself simultaneously causes ORF3a-Q57H, ORF3c-R36I, and ORF3d-E14* (*ORF3d*-LOF), its spread seems to be linked with RdRp-P323L and Spike-D614G, a variant with predicted functional relevance (e.g. see *Korber et al., 2020*; *Plante et al., 2020*; *Yurkovetskiy et al., 2020*). Thus, it is possible that G25563U is neutral while a linked variant(s) is under selection. These observations highlight the necessity of empirically evaluating the effects of these mutations and their interactions. Lastly, because only five major polymorphisms are considered in this analysis (*Supplementary file 1*), it is possible that the spread of *ORF3d*-LOF or other haplotypes was further assisted or hindered by subsequent mutations.

Our study has several limitations. First, we were not able to confirm the translation of ORF3d using mass spectrometry (MS). This may be due to several reasons: (1) ORF3d may be expressed at low levels; (2) ORF3d is short, and tryptic digestion of ORF3d generates only two peptides potentially detectable by MS; (3) the tryptic peptides derived from ORF3d may not be amenable to detection by MS even under the best possible conditions, as suggested by its relatively low MS intensity even in an overexpression experiment ('ORF3b' in *Gordon et al., 2020*); and (4) hitherto unknown post-translational modifications of ORF3d could prohibit detection. Other possibilities for validation of ORF3d include MS with other virus samples; affinity purification MS; fluorescent tagging and cell imaging; Western blotting; and the sequencing of additional genomes in this virus species, which would potentiate more powerful tests of purifying selection and a better understanding of the history and origin of *ORF3d*. With respect to between-host diversity, we focused on consensus-level sequence data; however, this approach can miss important variation (*Holmes, 2009*), stressing the importance of deeply sequenced within-host samples using technology appropriate for calling within-host variants (*Grubaugh et al., 2019*). As we use Wuhan-Hu-1 for reference-based read mapping and remove duplicate reads as possible PCR artifacts, reference bias (*Degner et al., 2009*) or bias against natural duplicates at high-coverage loci (*Zhou et al., 2014*) could potentially affect our within-host results. Additionally, we detected natural selection using so-called 'counting' methods that examine all pairwise comparisons between or within specific groups of sequences, which may have less power than methods that trace changes over a phylogeny. However, this approach is robust to errors in phylogenetic and ancestral sequence reconstruction, and to artifacts due to linkage or recombination (*Hughes et al., 2006*; *Nelson and Hughes, 2015*). Additionally, although our method for measuring selection in OLGs does not explicitly account for mutation bias, benchmarking suggests inference of purifying selection is conservative (*Nelson et al., 2020a*). Finally, given multiple recombination breakpoints in *ORF3a* (*Boni et al., 2020*) and the relative paucity of sequence data for viruses closely related to SARS-CoV-2, our analysis could not differentiate between convergence, recombination, or recurrent loss in the origin of *ORF3d*.

In conclusion, OLGs are an important part of viral biology and deserve more attention. We document several lines of evidence for the expression and functionality of a novel OLG in SARS-CoV-2, *ORF3d*, and compare it to other hypothesized OLG candidates in *ORF3a*. Finally, as a resource for future studies, we provide a detailed annotation of the SARS-CoV-2 genome and highlight mutations of potential relevance to the within- and between-host evolution of SARS-CoV-2.

## Materials and methods

### Data and software

Supplementary scripts and documentation are freely available on GitHub at https://github.com/chasewnelson/SARS-CoV-2-ORF3d (*Nelson et al., 2020b*; copy archived at swh:1:rev:469689991e7dbce2be4c4f618584304d91841c49). Supplementary data not included in main figure source data are freely available on Zenodo at https://zenodo.org/record/4052729.

## Genomic features and coordinates

All genome coordinates are given with respect to the Wuhan-Hu-1 reference sequence (NCBI: NC_045512.2; GISAID: EPI_ISL_402125) unless otherwise noted. SARS-CoV (SARS-CoV-1) genome coordinates are given with respect to the Tor2 reference sequence (NC_004718.3). SARS-CoV-2 Uniprot peptides were obtained from https://viralzone.expasy.org/8996, where *ORF9c* is currently referred to as *ORF14*. Nucleotide sequences were translated using R::Biostrings (*Lawrence et al., 2013*), Biopython (*Cock et al., 2009*), or SNPGenie (*Nelson et al., 2015*). Alignments were viewed and edited in AliView v1.20 (*Larsson, 2014*). To identify overlapping genes (OLGs) using the codon permutation method of *Schlub et al., 2018*, all 12 open reading frames (ORFs) annotated in Wuhan-Hu-1 were used as a reference (*ORF1a*, *ORF1b*, *S*, *ORF3a*, *E*, *M*, *ORF6*, *ORF7a*, *ORF7b*, *ORF8*, *N*, and *ORF10*) (*Figure 1*; *Supplementary file 1*). Only forward-strand (same-strand; sense-sense) OLGs were considered in the codon permutation analysis.

## SARS-CoV-2 genome data, alignments, and between-host analyses

SARS-CoV-2 genome sequences were obtained from GISAID on April 10, 2020 (*Supplementary file 2*). Whole genomes were aligned using MAFFT v7.455 (*Katoh and Standley, 2013*), and subsequently discarded if they contained internal gaps (-) located >900 nt from either terminus, a distance sufficient to exclude sequences with insertions or deletions (indels) in protein-coding regions. A total of 3978 sequences passed these filtering criteria, listed in *Supplementary file 1*. Coding regions were identified using exact or partial homology to SARS-CoV-2 or SARS-CoV annotations. To quantify the diversity and evenness of sample locations, we quantified their entropy as $-\sum p*\ln(p)$, where $p$ is the number of distinct (unique) locations or countries reported for a given window (*Ewens and Grant, 2001*; *Supplementary file 1*).

## *Severe acute respiratory syndrome-related coronavirus* genome data and alignments

*Severe acute respiratory syndrome-related coronavirus* genome IDs were obtained from *Lam et al., 2020* and downloaded from GenBank or GISAID. Wuhan-Hu-1 was used to represent SARS-CoV-2. Unless otherwise noted, isolates GX/P5L (NCBI: MT040335.1; GISAID: EPI_ISL_410540) and GD/1 (GISAID: EPI_ISL_410721) were used to represent pangolin-COVs; GD/1 was chosen as a representative because the other Guangdong (GD) sequence lacks the *S* gene and contains 27.76% Ns, while GX/P5L was chosen because it is one of two high-coverage Guangxi (GX) sequences derived from lung tissue that also contains no undetermined nucleotides (Ns). Other sequences were excluded if they lacked an annotated *ORF1ab* with a ribosomal slippage, or contained a frameshift indel in any gene (*Supplementary file 1*), leaving 21 sequences for analysis (*Supplementary file 1*).

To produce whole-genome alignments, we first aligned all genome sequences using MAFFT. Then, coding regions were identified using exact or partial sequence identity to SARS-CoV-2 or SARS-CoV annotations, translated, and individually aligned at the amino acid level using ProbCons v1.12 (*Do et al., 2005*). In the case of OLGs, the longest (reference) protein was used (e.g. N rather than ORF9b or ORF9c). Amino acid alignments were then imposed on the coding sequence of each gene using PAL2NAL v14 (*Suyama et al., 2006*) to maintain intact codons. Finally, whole genomes were manually shifted to match the individual codon alignments in AliView. Codon breaks were preferentially resolved to align S/Q/T at 3337–3339, and L/T/I at 3343–3345. This preserved all nucleotides of each genome while concurrently producing codon-aware alignments. The alignment is available in the supplementary data as SARS-related-CoV_ALN.fasta, where pangolin-CoV GD/1 has been masked (Ns) because it is only available from GISAID (i.e. permission is required for data access).

Phylogenetic relationships among isolates were explored using maximum likelihood phylogenetic inference, as implemented in IQ-tree (*Nguyen et al., 2015*), using the generalized time-reversible (GTR; *Tavaré, 1986*) substitution model combined with the FreeRate model (*Soubrier et al., 2012*) to account for among-site rate heterogeneity. Trees were rooted a priori following *Lam et al., 2020*.

## Proteomics analysis

We used MaxQuant (*Cox and Mann, 2008*) to re-analyze five publicly available SARS-CoV-2 mass spectrometry (MS) datasets: *Bezstarosti et al., 2020* (PRIDE accession PXD018760); *Bojkova et al.,*

*2020* (PXD017710); *Davidson et al., 2020* (PXD018241); PRIDE Project PXD018581; and *Zecha et al., 2020* (PXD019645). However, peptide spectrum matches for ORF3d (possible tryptic peptides CTSCCFSER and FQNHNPQK) did not pass our 1% false-discovery threshold in any of these datasets, and ORF3d-2 (*Table 1*) does not encode any peptides detectable by this method. ORF3c, ORF9c, and ORF10 could also not be reliably detected. For peptides that were successfully detected in the datasets of *Bezstarosti et al., 2020* and *Davidson et al., 2020*, protein concentrations were estimated using intensity-based absolute quantification (iBAQ) (*Schwanhäusser et al., 2011*). iBAQ values were computed using the Max-Quant software (*Tyanova et al., 2016*) as the sum of all peptide intensities per protein divided by the number of theoretical peptides per protein. Thus, iBAQ values are proportional estimates of the molar protein quantity of a protein in a given sample, allowing relative quantitative comparisons of protein expression levels.

## Ribosome profiling analysis

The 16 ribosome profiling (Ribo-seq) datasets of *Finkel et al., 2020* using SARS-CoV-2 infected Vero E6 cells were downloaded from the Sequence Read Archive (accession numbers SRR11713354-SRR11713369). These samples comprise four treatments for ribosome stalling: (1) lactimidomycin (LTM) or (2) harringtonine (Harr), which are biased towards stalling at translation initiation sites; (3) cyclohexamide (CHX), which stalls along a gene during active translation; and (4) mRNA, which serves as a control (i.e. total RNA content rather than ribosome footprints). Two time points are represented: (1) 5 hr and (2) 24 hr post-infection (hpi), labeled '05 hr' and '4 hr', respectively, in the SRA data. The FASTQ format reads were mapped to Wuhan-Hu-1 using Bowtie2 local alignment (*Langmead et al., 2019*), with a seed length of 20 and up to one mismatch allowed, after substituting the isolate's mutations listed in *Finkel et al., 2020* into the reference genome. Mapped reads were then classified by read length and trimmed to their first (5′-end) nucleotide for downstream analyses.

## NetMHCpan T cell epitope analysis

For predicted MHC class I binding, viral protein sequences were analyzed in 9 amino acid (aa) substrings using NetMHCpan4.0 (*Jurtz et al., 2017*). Twelve (12) representative HLA class I alleles (*Sidney et al., 2008*) were tested: HLA-A*01:01 (A1), HLA-A*02:01 (A2), HLA-A*03:01 (A3), HLA-A*24:02 (A24), HLA-A*26:01 (A26), HLA-B*07:02 (B7), HLA-B*08:01 (B8), HLA-B*27:05 (B27), HLA-B*39:01 (B39), HLA-B*40:01 (B44), HLA-B*58:01 (B58), and HLA-B*15:01 (B62). NetMHCpan4.0 returns percentile ranks that characterize a peptide's likelihood of antigen presentation compared to a set of random natural peptides. We employed the suggested threshold of 2% to determine potential presented peptides, and 0.5% to identify strong MHC class I binders. Both strong and weak binders were considered predicted epitopes. For predicted MHC class II binding, viral protein sequences were analyzed in 15 aa substrings using NetMHCIIpan4.0 (*Reynisson et al., 2020*). Twenty-six (26) representative HLA class II alleles (*Greenbaum et al., 2011*; *Paul et al., 2016*) were tested: HLA-DPA1*0201-DPB1*0401, HLA-DPA1*0103-DPB1*0201, HLA-DPA1*0201-DPB1*0101, HLA-DPA1*0201-DPB1*0501, HLA-DPA1*0301-DPB1*0402, HLA-DQA1*0101-DQB1*0501, HLA-DQA1*0102-DQB1*0602, HLA-DQA1*0301-DQB1*0302, HLA-DQA1*0401-DQB1*0402, HLA-DQA1*0501-DQB1*0201, HLA-DQA1*0501-DQB1*0301, DRB1*0101, DRB1*0301, DRB1*0401, DRB1*0405, DRB1*0701, DRB1*0802, DRB1*0901, DRB1*1101, DRB1*1201, DRB1*1302, DRB1*1501, DRB3*0101, DRB3*0202, DRB4*0101, and DRB5*0101. NetMHCIIpan4.0 returns percentile ranks based on an eluted ligand prediction score. We employed the suggested threshold of 10% to determine potential presented peptides, and 2% to identify strong MHC class II binders. Both strong and weak binders were considered predicted epitopes.

## Hydrophobicity profile analysis

The sequence of *ORF3a* was translated in all three forward-strand reading frames and uploaded to the VOLPES server (http://volpes.univie.ac.at/; *Bartonek and Zagrovic, 2019*) to determine hydrophobicity profiles using the unitless scale 'FAC1' (Factor 1; *Atchley et al., 2005*) with a window size of 25 aa. The correlations between the profiles for the reading frames were calculated for peptides encoded by the following subregions of *ORF3a*, from 5′ to 3′: (1) '*ORF3a*', codons not overlapping any known or hypothesized overlapping genes; (2) '*ORF3a/ORF3c*', encoding residues of both

ORF3a and ORF3c; (3) '*ORF3a/ORF3c/ORF3d*', encoding residues of ORF3a, ORF3c, and ORF3d; (4) '*ORF3a/ORF3d*', encoding residues of both ORF3a and ORF3d; and (5) '*ORF3a/ORF3b*', encoding residues of both ORF3a and ORF3b. Spearman's rank correlation was computed using stats::cor.test with method='spearman' in R, treating mean 25-residue window values as the observational unit.

## Statistics and tests of natural selection

All p-values reported in this study are two-tailed. Statistical and data analyses and visualization were carried out in R v3.5.2 (*R Development Core Team, 2018*) (libraries: boot, feather, ggrepel, patchwork, RColorBrewer, scales, tidyverse), Python (BioPython, pandas) (*McKinney, 2010*), Microsoft Excel, Google Sheets, and PowerPoint. Colors were explored using Coolors (https://coolors.co). Copyright-free images of a bat, human, and pangolin were obtained from Pixabay (https://pixabay.com). All $d_N/d_S$ or $\pi_N/\pi_S$ ratios were estimated for non-OLG regions using SNPGenie scripts snpgenie.pl or snpgenie_within_group.pl (*Nelson et al., 2015*; https://github.com/chasewnelson/SNPGenie), and for OLG regions using OLGenie script OLGenie.pl (*Nelson et al., 2020a*; https://github.com/chasewnelson/OLGenie). All OLG $d_N/d_S$ ($\pi_N/\pi_S$) estimates refer to $d_{NN}/d_{SN}$ ($\pi_{NN}/\pi_{SN}$) for the reference frame and $d_{NN}/d_{NS}$ ($\pi_{NN}/\pi_{NS}$) for the alternate frame (ss12 or ss13), as described in *Nelson et al., 2020a*, because the number of SS (synonymous/synonymous) sites was insufficient to estimate $d_{SS}$ ($\pi_{SS}$). The null hypothesis that $d_N$-$d_S$ = 0 ($\pi_N$-$\pi_S$ = 0) was evaluated using both $Z$ and achieved significance level (ASL) tests (*Nei and Kumar, 2000*) with 10,000 and 1000 bootstrap replicates for genes and sliding windows, respectively, using individual codons (alignment columns) as the resampling unit (*Nei and Kumar, 2000*). The main text reports only $Z$-test results, because this test was used to benchmark OLGenie (*Nelson et al., 2020a*) and appears to be more conservative. For ASL, p-values of 0 were reported as the lowest non-zero value possible given the number of bootstrap replicates. Benjamini-Hochberg (*Benjamini and Hochberg, 1995*) or Benjamini-Yekutieli (*Benjamini and Yekutieli, 2001*) false-discovery rate corrections (Q-values) were used for genes (independent regions) and sliding windows (contiguous overlapping regions), respectively.

## Between-species analyses

Because uncorrected $d$ values >0.1 were observed in between-taxa comparisons, a Jukes-Cantor correction (*Jukes and Cantor, 1969*) was applied to $d_N$ and $d_S$ estimates. For each ORF, sequences were only used to estimate $d_N/d_S$ if a complete, intact ORF (no STOPs) was present. Most notably, for *ORF3b*, only the region corresponding to the first ORF in SARS-CoV-2 was analyzed, and these sites were also considered an OLG region of *ORF3a*. The remaining 3'-proximal portion of *ORF3a* (codons 165–276) was considered a non-OLG region, and only genomes lacking the full-length *ORF3b* (i.e. those with mid-sequence STOP codons) were included in these estimates. Similarly, for *ORF3d*, only SARS-CoV-2 and pangolin-CoV GX/P5L were analyzed for *Figure 5*. For this analysis, *ORF3a* codon 71 in SARS-CoV-2 (CTA) differed by two nucleotides from the codon in pangolin-CoV GX/P5L (TTT), resulting in a multi-hit approximation being employed by the OLGenie method (*Nelson et al., 2020a*). Thus, for greater accuracy, we instead estimated changes in this codon using the method of *Wei and Zhang, 2015* across two mutational pathways, which yields 0.5 nonsynonymous/nonsynonymous, 0.5 nonsynonymous/synonymous, and 1.0 synonymous/nonsynonymous changes. The region occupied by the triple overlap of *ORF3a/ORF3c/ORF3d* (*ORF3a* codons 44–64; *Table 1*) was excluded from analysis for all three genes. The following codons (or their homologous positions) were also excluded from all analyses: all codons occupying a non-OLG/OLG boundary; codons 4460–4466 of *ORF1ab*, which constitute either *nsp11* or *nsp12* depending on a ribosomal frameshift; codons 1–13 of *E*, which overlap *ORF3b* in some genomes; codons 62–64 of *ORF6*, which follow a premature STOP in some genomes; codons 122–124 of *ORF7a* and 1–3 of *ORF7b*, which overlap each other; and codons 72–74 of *ORF9c*, which follow a premature STOP in some genomes.

## Cumulative haplotype frequencies

We defined haplotypes along the mutational path to EP+1+LOF using all five high-derived allele frequency (DAF) mutations from Wuhan-Hu-1 to *ORF3d*-LOF. Subsequent mutations after *ORF3d*-LOF were ignored in the haplotype analysis. Samples with missing data at any of the five loci were also ignored. We calculated the cumulative frequency of each haplotype in Germany (where the EP haplotype is a documented founder) and five other countries with the most abundant samples at the

time of data accession. Cumulative frequencies were calculated as the total number of occurrences of each haplotype collected on or before each day, divided by the total number of samples from the same country. Countries were subsequently divided into early founders and late founders to investigate founder effects, where early founder countries tend to have several samples from January, and late founder countries tend to have samples collected only after mid-February.

## Within-host diversity

For within-host analyses, we obtained $n = 401$ high-depth (at least 50-fold mean coverage) human SARS-CoV-2 samples from the Sequence Read Archive (listed in *Supplementary file 1*). Only Illumina samples were used, as some Nanopore samples exhibit apparent systematic bias in calling putative intrahost SNPs, and this technology has also been shown to be unsuitable for intra-host analysis (*Grubaugh et al., 2019*). Reads were trimmed with BBTools BBDUK (*Bushnell, 2017*) and mapped against the Wuhan-Hu-1 reference sequence using Bowtie2 (*Langmead and Salzberg, 2012*) with local alignment, seed length 20, and up to one mismatch. SNPs were called from mapped reads using the LoFreq (*Wilm et al., 2012*) variant caller, requiring both sequencing quality and MAPQ to be $\geq 30$. Only single-end or the first of paired-end reads were used. Within-host variants were dynamically filtered based on each site's coverage using a binomial cutoff to ensure a false-discovery rate of $\leq 1$ across our entire study (401 samples), assuming a mean sequencing error rate of 0.2% (*Schirmer et al., 2016*).

To estimate $\pi$, numbers of nonsynonymous and synonymous differences and sites were first calculated individually for all genes in each of the 401 samples using SNPGenie (*Nelson et al., 2015*; snpgenie.pl, https://github.com/chasewnelson/SNPGenie). OLG regions were then analyzed separately using OLGenie (*Nelson et al., 2020a*; OLGenie.pl, https://github.com/chasewnelson/OLGenie). Because OLGenie requires a multiple sequence alignment as input, a pseudo-alignment of 1000 sequences was constructed for each OLG region in each sample by randomly substituting single nucleotide variants according to their within-host frequencies into the Wuhan-Hu-1 reference genome. For non-OLG and OLG regions alike, average within-host numbers of differences and sites were calculated for each codon by taking the mean across all samples. For example, if a particular codon contained nonsynonymous differences in two of 401 samples, with the two samples exhibiting mean numbers of 0.01 and 0.002 pairwise differences per site, this codon was considered to exhibit a mean of (0.01+0.002)/401 = 0.0000299 pairwise differences per site across all samples. These codon means were then treated as independent units of observation during bootstrapping.

Pangolin samples refer to Sequence Read Archive records SRR11093266, SRR11093267, SRR11093268, SRR11093269, SRR11093270, and SRR11093271. Only 179 single nucleotide variants could be called prior to our FDR filtering, and samples SRR11093271 and SRR11093270 were discarded entirely due to low mapping quality. We also note that after our quality filtering, four samples contain consensus alleles that do not match their reference sequence (available from GISAID): P1E, P4L, P5E, and P5L (*Supplementary file 1*).

## Within-host recurrent mutations analyses

We assume that each host was infected by a single genotype. Under this assumption, the minor allele at each segregating site within-host is either due to genotyping or sequencing artifacts, or new mutations. Because there are very few loci with high-frequency derived alleles between hosts, and because Wuhan-Hu-1 is used as the reference in read mapping, we here only consider within-host mutations that differ from this reference background. There are four possible bases at each locus (A, C, G, and U), and three possible mutational changes away from the reference. For each locus, we calculated the number of samples matching the reference allele as $N = N_1 + N_2$, where $N_1$ is the number of samples in which the Wuhan-Hu-1 reference allele is the only observed allele (fixed), and $N_2$ is the number of samples in which the site is polymorphic but the reference allele is still the major (most common) allele. Given $N_2$, we further determined the number of samples carrying each of the three possible non-reference alleles as $N_A$, $N_C$, $N_G$, and $N_U$. For example, if the reference allele was U, we calculated $p_A$, $p_C$, and $p_G$, where $p_{All} = p_A + p_C + p_G$. If A was an observed non-reference allele, we calculated the frequency of A among samples as $p_A = N_A/N$. Thus, a larger frequency indicates the derived allele is observed in a higher proportion of samples (*Figure 8*). The within-host derived allele frequency (DAF) for each sample was estimated as the total

number of reads matching the observed minor allele divided by the total number of reads mapped to the locus. If all reads matched the reference allele, then DAF = 0. Five mutations (four nonsynonymous) occur in $\geq$10% of samples (after rounding to the nearest percent), with their DAFs plotted in *Figure 8—figure supplement 1*. For this analysis, we did not apply the per-site FDR cutoff, thus a DAF = 0 is equivalent to the absence of mapped reads with the mutation, after reads are filtered by sequence quality, mapping quality, and LoFreq's default significance threshold (p=0.01).

## Acknowledgements

This work was supported by a Postdoctoral Research Fellowship from Academia Sinica (to CWN under PI Wen-Hsiung Li); funding from the Bavarian State Government and National Philanthropic Trust (to Z.A. under PI Siegfried Scherer); NSF IOS grants #1755370 and #1758800 (to S-OK); and the University of Wisconsin-Madison John D MacArthur Professorship Chair (to TLG). Copyright-free images were obtained from Pixabay. The authors thank the GISAID platform and the originating and submitting laboratories who kindly uploaded SARS-CoV-2 sequences to the GISAID EpiCov Database for public access (see *Supplementary file 2*, GISAID Acknowledgments Table). The authors thank Maciej F Boni, Reed A Cartwright, John Flynn, Kyle Friend, Dan Graur, Robert S Harbert, Cheryl Hayashi, David G Karlin, Niloufar Kavian, Kin-Hang (Raven) Kok, Wen-Hsiung Li, Meiyeh Lu, David A Matthews, Lisa Mirabello, Apurva Narechania, Felix Li Jin, and attendees of the UC Berkeley popgen journal club for useful information and discussion; Andrew E Firth, Alexander Gorbalenya, Irwin Jungreis, Manolis Kellis, Raven Kok, Angelo Pavesi, Kei Sato, Manuela Sironi, and Noam Stern-Ginossar for an invaluable discussion regarding standardizing nomenclature; Angelo Pavesi for pointing out that *ORF3d*-LOF causes a nonsynonymous change in *ORF3c*; Helen Piontkivska, Patricia Wittkopp, Antonis Rokas, and one anonymous reviewer for critical suggestions; Ming-Hsueh Lin for immense feedback on figures; and special thanks to Priya Moorjani, Jacob Tennessen, Montgomery Slatkin, Yun S Song, Jianzhi George Zhang, Xueying Li, Hongxiang Zheng, Qinqin Yu, Meredith Yeager, and Michael Dean for commenting on earlier drafts of this manuscript.

## Additional information

### Funding

| Funder | Grant reference number | Author |
| --- | --- | --- |
| Academia Sinica | Postdoctoral Research Fellowship | Chase W Nelson |
| National Philanthropic Trust | Grant | Zachary Ardern |
| University of Wisconsin-Madison | John D MacArthur Professorship Chair | Tony L Goldberg |
| National Science Foundation | IOS grants #1755370 and #1758800 | Sergios-Orestis Kolokotronis |

The funders had no role in study design, data collection and interpretation, or the decision to submit the work for publication.

### Author contributions

Chase W Nelson, Zachary Ardern, Conceptualization, Resources, Data curation, Software, Formal analysis, Supervision, Validation, Investigation, Visualization, Methodology, Writing - original draft, Project administration, Writing - review and editing; Tony L Goldberg, Conceptualization, Supervision, Funding acquisition, Methodology, Project administration, Writing - review and editing; Chen Meng, Formal analysis, Investigation, Methodology; Chen-Hao Kuo, Conceptualization, Software, Formal analysis, Validation, Investigation, Writing - review and editing; Christina Ludwig, Resources, Formal analysis, Supervision, Methodology; Sergios-Orestis Kolokotronis, Conceptualization, Formal analysis, Supervision, Funding acquisition, Investigation, Visualization, Methodology, Writing - review and editing; Xinzhu Wei, Conceptualization, Resources, Software, Formal analysis, Supervision,

Validation, Investigation, Visualization, Methodology, Writing - original draft, Project administration, Writing - review and editing

Author ORCIDs
Chase W Nelson https://orcid.org/0000-0001-6287-1598
Zachary Ardern https://orcid.org/0000-0002-9008-0897
Tony L Goldberg https://orcid.org/0000-0003-3962-4913
Chen Meng https://orcid.org/0000-0002-5968-6719
Chen-Hao Kuo https://orcid.org/0000-0002-8454-0615
Christina Ludwig http://orcid.org/0000-0002-6131-7322
Sergios-Orestis Kolokotronis https://orcid.org/0000-0003-3309-8465
Xinzhu Wei https://orcid.org/0000-0001-8184-7016

**Decision letter and Author response**
Decision letter https://doi.org/10.7554/eLife.59633.sa1
Author response https://doi.org/10.7554/eLife.59633.sa2

## Additional files

### Supplementary files
• Supplementary file 1. SARSCoV2_ORF3d_supplementary_tables.xlsx. Supplementary Tables; detailed descriptions given in the CONTENTS sheet.

• Supplementary file 2. gisaid_hcov-19_acknowledgement_table_2020_04_28_08.xls. List of laboratories submitting SARS-CoV-2 consensus-level genome sequences to the GISAID EpiCov Database for public access.

• Transparent reporting form

### Data availability
All data generated or analyzed during this study are included in the manuscript and supplement. Scripts and source data for all analyses and figures are provided on GitHub at https://github.com/chasewnelson/SARS-CoV-2-ORF3d (copy archived at https://archive.softwareheritage.org/swh:1:rev:469689991e7dbce2be4c4f618584304d91841c49) and Zenodo at https://zenodo.org/record/4052729. Samples SRR11713366, SRR11713367, SRR11713368, and SRR11713369 were used from the GSE149973 dataset. The EpiCoVTM database was accessed at https://www.gisaid.org/ and exact sequences used from this database (prior to filtering) are listed in Supplementary file 2.

The following previously published datasets were used:

| Author(s) | Year | Dataset title | Dataset URL | Database and Identifier |
|---|---|---|---|---|
| Demmers J | 2020 | TARGETED PROTEOMICS FOR THE DETECTION OF SARS-COV-2 PROTEINS | https://www.ebi.ac.uk/pride/archive/projects/PXD018760 | PRIDE, PXD018760 |
| Klann K, Münch C | 2020 | Proteome and Translatome of SARS-CoV-2 infected cells | https://www.ebi.ac.uk/pride/archive/projects/PXD017710 | PRIDE, PXD017710 |
| Mari T, Selbach M | 2020 | Proteomics of SARS-CoV and SARS-CoV-2 infected cells | https://www.ebi.ac.uk/pride/archive/projects/PXD018581 | PRIDE, PXD018581 |
| Finkel Y, Mizrahi O, Nachshon A, Weingarten-Gabbay S | 2020 | Decoding SARS-CoV-2 coding capacity | https://www.ncbi.nlm.nih.gov/geo/query/acc.cgi?acc=GSE149973 | NCBI Gene Expression Omnibus, GSE149973 |
| Lee CY, Kuster B | 2020 | Data, reagents, assays and merits of proteomics for SARS-CoV-2 research and testing | https://www.ebi.ac.uk/pride/archive/projects/PXD019645 | PRIDE, PXD019645 |
| Matthews D | 2020 | Vero cells infected with SARS CoV 2 no quantitation slices 1-10 of 20 | https://doi.org/10.5281/zenodo.3722590 | Zenodo, 10.5281/zenodo.3722590 |

| Matthews D | 2020 | vero cells infected with SARS CoV2 slices 11-20 of 20 slices | https://doi.org/10.5281/zenodo.3722597 | Zenodo, 10.5281/zenodo.3722597 |
|---|---|---|---|---|

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
