## [Decision Letter]

**Acceptance summary:**

This study provides evidence for the existence of a novel gene in genomes of the SARS-CoV-2 virus that may be associated with the origin of the ongoing pandemic. The novel gene overlaps with other known genes in the SARS-CoV-2 genome and exhibits a high rate of evolutionary change. The study also offers a comparison between other overlapping ORFs identified in SARS-CoV-2 genomes, their distributions in other coronavirus species, and their rates of evolution. Given the confusion around the reported overlapping ORFs so far and the fact that these genes are quite understudied, this work is important in establishing the existence of a novel overlapping genes and will spur further analyses into its role in the SARS-CoV-2 pathogenesis.

**Decision letter after peer review:**

Thank you for submitting your article "A previously uncharacterized gene in SARS-CoV-2 and the origins of the COVID-19 pandemic" for consideration by *eLife*. Your article has been reviewed by two peer reviewers, and the evaluation has been overseen by a Reviewing Editor and Patricia Wittkopp as the Senior Editor. The following individual involved in review of your submission has agreed to reveal their identity: Helen Piontkivska (Reviewer #2).

The reviewers have discussed the reviews with one another and the Reviewing Editor has drafted this decision to help you prepare a revised submission.

Summary:

This manuscript describes a putative novel gene from the SARS-CoV-2 genomes that may be associated with the origin of the ongoing pandemic. While initially identified through in-silico genome sequence analysis, the authors offer additional evidence in support of the functionality of this overlapping ORF derived from Ribo-Seq data, based on the ribosomes' accumulation at the translation start site, as well as protein structure and mass spec analyses. They further examine the extent of sequence conservation of this putative ORF and the remainder of the genome across SARS-CoV-2 isolates, as well as in genomes of other coronaviruses. The results indicate while the majority of sites is subject to relaxed purifying selection, this younger overlapping *ORF3c* shows exhibits higher levels of nonsynonymous changes, which may potentially be linked to antibody-mediated immune response, among other factors. This is an interesting study that identifies a novel overlapping gene, *ORF3c*, which may have played a role in host jump. The study further offers a comparison between other overlapping ORFs identified in SARS-CoV-2 genomes and their distribution in other coronavirus species (Figure 1 and Table 1). Given the confusion around the reported overlapping ORFs so far and the fact that these genes are quite understudied, this work is important in establishing the existence of *ORF3c* and will spur further analyses into its role in the SARS-CoV-2 pathogenesis.

Essential revisions:

1) Title: The title is somewhat underwhelming, and can be revised to better reflect the major points of the study (e.g., emphasize novel overlapping ORF and its dynamic origin? Something like "Dynamically evolving novel overlapping gene as a factor in SARS-CoV-2 pandemic", or something along those lines).

2) The authors make an important point about correct annotation and description of novel/putative ORFs within RNA virus genomes. They bring together a wide variety of different data sources to suggest that *ORF3c* is real, expressed, a target of the immune system and potentially under natural selection. However, many of their arguments (particularly with regards to the population genomics of SARS-CoV-2 lineages/haplotypes) lean too heavily on arguments of natural selection on particular genotypes, when simpler explanations exist (drift, founder effects and the uneven impact of virus population control across and within countries). While they acknowledge these factors, the authors nevertheless are too quick to discount them. Please avoid "adaptationist" language in the absence of specific analyses inferring the action of selection and ensure that all alternative explanations, including ones that do not involve selection, are adequately and fairly discussed.

3) A significant weakness in this paper is the lack of good evidence that *ORF3c* is expressed at the protein level. We understand that the ORF is too small to be reliably detected by mass spec, but a Western blot could be used to solve this problem. Would it be possible to provide such data? If this is not possible, it is important to ensure that readers are aware of this shortcoming of this study.

4) This paper frequently refers to other publications which are not well-respected by other members of the genomics and virology community (the Forster PNAS paper; the Bhattacharyya 2020 spike paper and others discussing the role of the D/G spike mutant).

5) The MHC prediction analysis uses NetMHCpan, which predicts CD8^+^ T cell epitopes (presented by MHC class I), but does not consider MHC class II epitope presentation to CD4^+^ T cells, which have been shown to be important for SARS-CoV-2 infection and which bind to different peptides in different genomic regions compared to CD8^+^ T cells.

6) Figure 1 would benefit from having coordinates shown for all the overlapping ORFs (per Wuhan-Hu-1 reference, perhaps?)

7) Could a separate line that incorporates the ORFs listed in Table 1 be added to Figure 1? It may enhance the impact on erroneous homology in the third paragraph of the Discussion.

8) We got lost in Figure 5; perhaps, the amino acid sequences can be shown in some other way or in a separate figure/panel? It would be important to show the location of stop codons relative to their alignment positions in *ORF3c* and *3b*.

9) Discussion, end of third paragraph: this sentence isn't quite complete. S-iORF2 shows evidence of translation, and so what?

10) The manuscript has a nice list of limitations; however, we would have also liked to see a few suggestions on how the role of *ORF3c* could be validated, even if authors do not plan to pursue those themselves.

---

## [Author Response]

Essential revisions:1) Title: The title is somewhat underwhelming, and can be revised to better reflect the major points of the study (e.g., emphasize novel overlapping ORF and its dynamic origin? Something like "Dynamically evolving novel overlapping gene as a factor in SARS-CoV-2 pandemic", or something along those lines).

Thanks for this great suggestion. We have adopted the proposed title.

2) The authors make an important point about correct annotation and description of novel/putative ORFs within RNA virus genomes. They bring together a wide variety of different data sources to suggest that ORF3c is real, expressed, a target of the immune system and potentially under natural selection. However, many of their arguments (particularly with regards to the population genomics of SARS-CoV-2 lineages/haplotypes) lean too heavily on arguments of natural selection on particular genotypes, when simpler explanations exist (drift, founder effects and the uneven impact of virus population control across and within countries). While they acknowledge these factors, the authors nevertheless are too quick to discount them. Please avoid "adaptationist" language in the absence of specific analyses inferring the action of selection and ensure that all alternative explanations, including ones that do not involve selection, are adequately and fairly discussed.

We completely agree with the reviewers. We have reworked our arguments to avoid adaptationist language, and emphasize non-selective mechanisms throughout, especially the potential roles of drift and founder effects. Three specific examples follow.

First, in the section on “Between-host evolution and pandemic spread”, we have changed the text to read as follows:

“During the first months of the COVID-19 pandemic, ORF3d-LOF increased in frequency (Table S15 in Supplementary file 1) in multiple locations (Figure 5—figure supplement 2D; Table S16 in Supplementary file 1), making this mutation a candidate for natural selection on *ORF3a*, ORF3d, or both (Kosakovsky-Pond, 2020). However, temporal allele frequency trajectories (Figure 5—figure supplement 2D) and similar signals from phylogenetic branch tests may also be caused by founder effects or genetic drift, and are susceptible to ascertainment bias (e.g., preferential sequencing of imported infections and uneven geographic sampling) and stochastic error (e.g., small sample sizes).”

Second, in the same section, we have also tempered our language when discussing potential driver mutations, now writing:

“Thus, while founder effects and drift are plausible explanations, it is also worth investigating whether the early spread of ORF3d-LOF may have been caused by its linkage with another driver, either C14408U (+1 variant) or a subsequent variant(s) occurring on the EP+1+LOF background (Discussion).”

Finally, in the Discussion, we have removed any mention of viral spread, hospitalization rate, etc., and instead written a full section detailing non-adaptationist explanations of the quick spread of certain variants:

“The quick expansion of ORF3d-LOF (EP+1+LOF) and its haplotype (EP+1) during this pandemic is surprising, given that a founder effect would have favored other variants that arrived earlier. […] Thus, it is possible that G25563U is neutral while a linked variant(s) is under selection.”

3) A significant weakness in this paper is the lack of good evidence that ORF3c is expressed at the protein level. We understand that the ORF is too small to be reliably detected by mass spec, but a Western blot could be used to solve this problem. Would it be possible to provide such data? If this is not possible, it is important to ensure that readers are aware of this shortcoming of this study.

We agree with the reviewers that a western blot with the antibody to ORF3d (previously *ORF3c*) would be the best way of validating its protein product. Via personal communication with Raven Kok, we were made aware that their research group at the University of Hong Kong has raised antibodies for ORF3d for such an experiment. As our group lacks the resources and expertise for western blot, and since this effort is being made by other groups, we instead focused on providing more analytical evidence from ribosome profiling. We now added new ribosome profiling results using the datasets from 5 hours post infection from Finkel et al., 2020. These new results, although not directly proving the stable presence of ORF3d protein, strongly support translation of *ORF3d*. Specifically, in the section on “ORF3d molecular biology and expression”, we show that ORF3d fits a pattern observed for all annotated genes, i.e., a local maximum in ribosome profiling read depth at the ribosome P-site offset of -12 nt relative to its start site (Figure 2A), and validate this approach by showing that the depth of these peaks correlates strongly (r=0.89, p=0.0004, Spearman’s rank) with protein levels measured from mass spectrometry (Figure 2B). We also conduct a frame-based analysis, showing that stalled ribosomes are uniquely enriched in the frame of ORF3d at its precise locus (Figure 2C).

With respect to mass spectrometry (MS), we have specifically discussed the limitations of this method in three places in the manuscript. First, in the section “ORF3d molecular biology and expression”, we have added:

“This result [lack of detection of ORF3d] may reflect the limitations of MS for detecting proteins that are very short, weakly expressed under the specific conditions tested, or lack detectable peptides. Even the envelope protein E is not detected in some SARS-CoV-2 studies (Bojkova et al., 2020; Davidson et al., 2020), and the only evidence for *ORF3b* expression in SARS-CoV comes from Vero E6 cells (McBride and Fielding, 2012).”

Second, we have also added text discussing other possibilities for validation of ORF3d translation (see response #10). Finally, we have also added further explanation in the limitations paragraph of the Discussion:

“First, we were not able to confirm the translation of ORF3d using mass spectrometry (MS). This may be due to several reasons: (1) ORF3d is short, and tryptic digestion of ORF3d generates only two peptides potentially detectable by MS; (2) ORF3d may be expressed at low levels; (3) the tryptic peptides derived from ORF3d may not be amenable to detection by MS even under the best possible conditions, as suggested by its relatively low MS intensity even in an overexpression experiment (‘ORF3b’ in Gordon et al., 2020); and (4) hitherto unknown post-translational modifications of ORF3d could also prohibit detection.”

4) This paper frequently refers to other publications which are not well-respected by other members of the genomics and virology community (the Forster PNAS paper; the Bhattacharyya 2020 spike paper and others discussing the role of the D/G spike mutant).

We agree, and have removed references to these controversial papers, instead replacing both with more appropriate references, Korber et al., 2020, or both Korber et al., 2020, and Yurkovetskiy et al., 2020.

5) The MHC prediction analysis uses NetMHCpan, which predicts CD8^+^ T cell epitopes (presented by MHC class I), but does not consider MHC class II epitope presentation to CD4^+^ T cells, which have been shown to be important for SARS-CoV-2 infection and which bind to different peptides in different genomic regions compared to CD8^+^ T cells.

The reviewer raises an important point. We agree that MHC II is also critically important, particularly as antibody responses depend on the CD4^+^ T cell response. We have therefore completed a new MHC II analysis using NetMHCIIpan, now included in Figure 3A. Briefly, full-length ORF3d also has one of the lowest CD4^+^ T cell epitope densities, after *ORF3b*, *ORF7b*, and *ORF10*; moreover, the short isoform ORF3d-2 has no CD4^+^ T cell epitopes, a significant depletion compared to both short unannotated ORFs (p=0.001) and its own amino acid content (p=0.001) (see manuscript). For continuity with the reviewers’ requests to cover all OLGs (comments #6, #7, #8, and #10), we also examined the older and more constrained *ORF3c* (previously ORF3h/ORF3a*), finding that *ORF3c* has the highest observed CD8^+^ T cell epitope density. See the fully updated section “Protein sequence properties” and Figure 3A.

6) Figure 1 would benefit from having coordinates shown for all the overlapping ORFs (per Wuhan-Hu-1 reference, perhaps?)

This is a great suggestion. To address this in general (along with comments #6, #7, #8, and #10), we have now included all *ORF3a* OLGs in all our analyses throughout the manuscript for continuity. With respect to this specific comment and figure, we have (1) split Figure 1 into two panels, A (top) and B (bottom), with the Wuhan-Hu-1 coordinate system shown as x-axis in panel A; (2) explicitly marked out the coordinates of *ORF3a* and its OLGs *ORF3c*, *ORF3d*, and *ORF3b* next to where they appear in the diagram; and (3) provided source data and Tables S2 and S7 in Supplementary file 1, where specific coordinates for all genes can be found with respect to Wuhan-Hu-1 and our Severe acute respiratory syndrome-related coronavirus alignment, respectively.

7) Could a separate line that incorporates the ORFs listed in Table 1 be added to Figure 1? It may enhance the impact on erroneous homology in the third paragraph of the Discussion.

To address this, we have (1) implemented the changes discussed in response to comment #6; (2) added all OLGs from Table 1 (i.e., now including *ORF3c*) in Figure 1A; (3) drawn special attention to the OLGs in *ORF3a* by listing their coordinates in Figure 1A; and (4) made explicit reference to the updated version of Figure 4, which annotates the SARS-CoV-2 STOP codons in the region that is homologous to SARS-CoV *ORF3b*. Also see response #8.

8) We got lost in Figure 5; perhaps, the amino acid sequences can be shown in some other way or in a separate figure/panel? It would be important to show the location of stop codons relative to their alignment positions in ORF3c and 3b.

Thanks for this excellent suggestion. We have split the figure (now Figure 4) into two panels, with (A) showing the amino acid sequences and (B) showing the alignments. For *ORF3b*, we opted to replace the SARS-CoV amino acid sequence with that of SARS-CoV-2 (Wuhan-Hu-1), thereby emphasizing the presence of STOP codons in this ORF in the latter.

9) Discussion, end of third paragraph: this sentence isn't quite complete. S-iORF2 shows evidence of translation, and so what?

To address this point, we have introduced a new paragraph in the Discussion (“Although our study focuses on ORF3d, other OLGs have been proposed in SARS-CoV-2 and warrant investigation…”). We also briefly address the other overlapping genes, as requested by the reviewers in comments #6, #7, #8, and #10. This specific sentence in question now reads as follows:

“Within S, S-iORF2 (genome positions 21768-21863) also shows evidence of translation from ribosome profiling (Finkel et al., 2020), and between-host comparisons suggest purifying selection (πN/πS=0.22, p=0.0278) (Table 1). We therefore suggest that the SARS-CoV-2 genome could contain additional undocumented or new OLGs.”

10) The manuscript has a nice list of limitations; however, we would have also liked to see a few suggestions on how the role of ORF3c could be validated, even if authors do not plan to pursue those themselves.

This is a great suggestion. We have added the following text to the limitations paragraph in the Discussion:

“Other possibilities for validation of ORF3d include MS with other virus samples; affinity purification MS; fluorescent tagging and cell imaging; Western blotting; and the sequencing of additional genomes in this viral species, which would potentiate more powerful tests of purifying selection and a better understanding of the history and origin of ORF3d.”